# Unifying Formal Explanations: A Complexity-Theoretic Perspective

**Shahaf Bassan**[1*]**, Xuanxiang Huang**[2*] **, Guy Katz**[1]
The Hebrew University of Jerusalem[1], Nanyang Technological University[2]
shahaf.bassan@mail.huji.ac.il, xuanxiang.huang@ntu.edu.sg,
g.katz@mail.huji.ac.il

## Abstract

Previous work has explored the computational complexity of deriving two fundamental types of explanations for ML model predictions: (i) *sufficient reasons*, which are subsets of input features that, when fixed, determine a prediction, and (ii) *contrastive reasons*, which are subsets of input features that, when modified, alter a prediction. Prior studies have examined these explanations in different contexts, such as non-probabilistic versus probabilistic frameworks and local versus global settings. In this study, we introduce a unified framework for analyzing these explanations, demonstrating that they can all be characterized through the minimization of a unified probabilistic value function. We then prove that the complexity of these computations is influenced by three key properties of the value function: (i) *monotonicity*, (ii) *submodularity*, and (iii) *supermodularity* — which are three fundamental properties in *combinatorial optimization*. Our findings uncover some counterintuitive results regarding the nature of these properties within the explanation settings examined. For instance, although the *local* value functions do not exhibit monotonicity or submodularity/supermodularity whatsoever, we demonstrate that the *global* value functions do possess these properties. This distinction enables us to prove a series of novel polynomial-time results for computing various explanations with provable guarantees in the global explainability setting, across a range of ML models that span the interpretability spectrum, such as neural networks, decision trees, and tree ensembles. In contrast, we show that even highly simplified versions of these explanations become NP-hard to compute in the corresponding local explainability setting.

## 1 Introduction

Despite substantial progress in methods for explaining ML model decisions, the literature has consistently unfortunately found that many desirable explanation types with different provable guarantees are computationally hard to obtain (Barceló et al., 2020; Van den Broeck et al., 2022), with the difficulty typically worsening in complex or highly non-linear models (Barceló et al., 2020; Adolfi et al., 2025; 2024). As a result, the *computational complexity* of obtaining explanations has become a central theoretical focus, with many recent works aiming to chart which types of explanations can be efficiently obtained for different kinds of models, and which remain out of reach (Barceló et al., 2020; Wäldchen et al., 2021; Arenas et al., 2022; 2023; Marzouk & De La Higuera, 2024; Ordyniak et al., 2023; Laber, 2024; Bhattacharjee & Luxburg, 2024; Blanc et al., 2021; 2022).

**From sufficient to contrastive reasons.** Among studies on the computational complexity of generating explanations, two fundamental types of explanations for ML models were extensively examined: (i) *sufficient reasons* and (ii) *contrastive reasons* (Barceló et al., 2020; Arenas et al., 2022; Audemard et al., 2022a; Arenas et al., 2021; Barceló et al., 2025; Marques-Silva & Ignatiev, 2022; Ignatiev et al., 2020b; Darwiche & Hirth, 2020). A *sufficient reason* is a subset of input features $S$ such that when these features are fixed to specific values, the model's prediction remains unchanged, regardless of the values assigned to the complementary set $\overline{S}$. A *contrastive reason* is a subset of input features $S$ such that modifying these features leads to a change in the model's prediction.

---
[*]Equal contribution.

Unlike additive attribution methods, which allocate importance scores across features but are often hard for humans to interpret (Kumar et al., 2020) or lack actionability (Bilodeau et al., 2024), sufficient and contrastive reasons provide *discrete, condition-like explanations* that directly answer "what is enough to justify this prediction?" or "what must change to flip it?". Their intuitive nature has given them a central role in many classic XAI methods (Ribeiro et al., 2018; Carter et al., 2019; Ignatiev et al., 2019; Dhurandhar et al., 2018), shown greater effectiveness in improving human prediction over additive models (Ribeiro et al., 2018; Yin & Neubig, 2022; Dhurandhar et al., 2018), and proved useful for downstream tasks such as bias detection (Balkir et al., 2022; Carter et al., 2021; La Malfa et al., 2021; Muthukumar et al., 2018), model debugging (Jacovi et al., 2021), and anomaly detection (Davidson et al., 2025). A well-established principle further holds that *smaller* sufficient and contrastive explanations enhance interpretability, making *minimality* a central guarantee of interest (Ribeiro et al., 2018; Lopardo et al., 2023; Carter et al., 2019; Barceló et al., 2020; Arenas et al., 2022; Blanc et al., 2021; Wäldchen et al., 2021).

Barceló et al. (2020) conducted one of the earliest studies on the complexity of deriving sufficient and contrastive reasons. Their work established that finding the minimal-sized sufficient reason for a decision tree is NP-hard, with the complexity further increasing to $\Sigma_2^P$-hard for neural networks. In the case of contrastive reasons, they demonstrated that computing the smallest possible explanation is solvable in polynomial time for decision trees but becomes NP-Hard for neural networks. Similar hardness results were later shown to hold for tree ensembles as well (Izza & Marques-Silva, 2021; Ordyniak et al., 2024; Audemard et al., 2022b).

**From non-probabilistic to probabilistic explanations.** A common criticism of the classic definition of sufficient and contrastive reasons is their rigidity and lack of flexibility, as they hold in an absolute sense over entire domains and can thus lead to excessively large or uninformative explanations (Ignatiev et al., 2019; 2020a; Arenas et al., 2022; Wäldchen et al., 2021). To address these limitations, the literature has shifted towards a more general definition that incorporates a *probabilistic* perspective on sufficient and contrastive reasons (Wäldchen et al., 2021; Arenas et al., 2022; Blanc et al., 2021; Izza et al., 2023; Xue et al., 2023). Under this framework, the goal is to identify subsets of input features that influence a prediction with a probability exceeding a given threshold $\delta$.

Wäldchen et al. (2021) were the first to study the complexity of probabilistic sufficient reasons, showing that for CNF classifiers the problem is NP$^{\mathbf{PP}}$-hard. This hardness extends to tree ensembles and neural networks (Barceló et al., 2020; Ordyniak et al., 2023). A central theoretical insight in the probabilistic setting is the *lack of monotonicity* in the probability function (Arenas et al., 2022; Izza et al., 2023), which makes even *subset minimal* explanations computationally hard. Strikingly, (Arenas et al., 2022) show that finding a subset minimal probabilistic sufficient reason is NP-hard even for decision trees — unlike in the non-probabilistic case, where such explanations are computable in polynomial time (Huang et al., 2021; Barceló et al., 2020).

**From local to global explanations.** In a more recent study, Bassan et al. (2024) extend the complexity analysis from the *local* (non-probabilistic) setting — where explanations are tied to individual predictions — to the *global* (non-probabilistic) setting, which seeks sufficient or contrastive reasons over entire domains. However, as with other non-probabilistic methods (Ignatiev et al., 2019; Barceló et al., 2020; Arenas et al., 2021; Darwiche & Hirth, 2020), the criteria are extremely strict — arguably even more so in the global setting than in the local one. For instance, any feature excluded from a global sufficient reason is deemed strictly redundant (Bassan et al., 2024), often making the explanation span nearly all input features and thus less informative.

OUR CONTRIBUTIONS

1. We unify previous explanation computation problems — including sufficient, contrastive, probabilistic, non-probabilistic, as well as local and global — into one framework, described as a minimization task over a unified value function. We then identify three fundamental properties of the value function that significantly impact the complexity of this task: (a) *monotonicity*, (b) *supermodularity*. and (c) *submodularity*.

2. Interestingly, we show that these properties behave in *strikingly different manners* depending on the structure of the value function. In particular, we demonstrate the surprising result that while the *local* value functions for both sufficient and contrastive reasons are *non-monotonic*, their *global* counterparts are *monotonic non-decreasing*. Moreover, we identify additional

intriguing properties unique to the global setting: the global sufficient value function is *supermodular*, whereas the global contrastive value function is *submodular* — in contrast to the local setting, where neither property holds.

3. We leverage these properties to derive new complexity results for explanation computation, revealing the intriguing finding that global explanations with guarantees can be computed efficiently, even though computing their local counterparts remains computationally hard. We demonstrate these findings across three widely used model types that span the interpretability spectrum: (i) neural networks, (ii) decision trees, and (iii) tree ensembles. First, we prove that while computing a subset-minimal local sufficient/contrastive probabilistic explanation is NP-hard even for decision trees (Arenas et al., 2022), its global counterpart can be computed in polynomial time. We further extend this result to any black-box model (including complex models such as neural networks and tree ensembles) when using empirical distributions. Specifically, we show that obtaining a subset-minimal global sufficient/contrastive explanation is achievable in polynomial time, whereas the local version remains NP-hard for these models.

4. Finally, we present an even stronger complexity result by exploiting the submodular and supermodular properties of the value functions in the global setting — properties that do not hold in the local case. Specifically, we show that it is possible to achieve provable *constant-factor* approximation guarantees for computing cardinally minimal global explanations — even for complex models like neural networks or tree ensembles — when the empirical distribution is fixed. In sharp contrast, we establish strong inapproximability results for the local setting, demonstrating that no *bounded* approximation is possible, even under very simple assumptions.

Owing to space constraints, we provide an overview of our main theorems and corollaries in the main text, with full proofs deferred to the appendix.

## 2 PRELIMINARIES

**Setting.** We consider an input space of dimension $n \in \mathbb{N}$, with input vectors $\mathbf{x} := (\mathbf{x}_1, \mathbf{x}_2, \ldots, \mathbf{x}_n)$. Each coordinate $\mathbf{x}_i$ may take values from its corresponding domain $\mathcal{X}_i$, which can be either discrete or continuous. The full input feature space is therefore $\mathbb{F} := \mathcal{X}_1 \times \mathcal{X}_2 \times \ldots \times \mathcal{X}_n$. We consider classification models $f : \mathbb{F} \to [c]$ where $c \in \mathbb{N}$ is the number of classes. Moreover, we consider a *generic* distribution $\mathcal{D} : \mathbb{F} \to [0, 1]$ over the input space. In many settings, however, accurately approximating $\mathcal{D}$ is computationally infeasible. A natural alternative is to instead work with a fixed empirical dataset $\mathbf{D} := \{\mathbf{z}^1, \mathbf{z}^2, \ldots, \mathbf{z}^{|\mathbf{D}|}\}$, which serves as a practical proxy for the "true" distribution, for instance, by taking $\mathbf{D}$ as a sampled subset from the available training data.

The explanations that we study are either local or global. In the local case, the explanations target a specific prediction $\mathbf{x} \in \mathbb{F}$, providing a form of reasoning for why the model predicted $f(\mathbf{x})$ for that instance. In the global case, explanations aim to reflect the general reasoning behind the behavior of $f$ across a wider region of the input space, independent of any specific $\mathbf{x}$, and to characterize its overall decision-making logic.

**Models.** While many results presented in this work are general (e.g., inherent properties of value functions), we also provide some model-specific complexity results for widely-used ML models. To broadly address the interpretability spectrum, we chose to analyze models ranging from those typically considered "black-box" to those commonly regarded as "interpretable". Specifically, we focus on: (i) decision trees, (ii) neural networks (all architectures at least as expressive as feed-forward ReLU networks), and (iii) tree ensembles, including majority-voting ensembles (e.g., random forests) and weighted-voting ensembles (e.g., XGBoost). Formal definitions are provided in the Appendix.

**Distributions.** We emphasize that in probabilistic explanation settings, the complexity can vary significantly with the input distribution $\mathcal{D}$. Particularly, we focus on three distribution types: (i) *general distributions*, which make no specific assumptions over $\mathcal{D}$ and thus encompass all possible distributions. We use these distributions mainly in proofs of properties that hold universally; (ii) *empirical distributions*, which include all distributions derived from the finite dataset $\mathbf{D}$ — an approach commonly employed in XAI (Lundberg & Lee, 2017; Van den Broeck et al., 2022); and (iii) *independent distributions*, which assume that features in $\mathcal{D}$ are mutually independent — another widely adopted

assumption in XAI literature (Arenas et al., 2022; 2023; Lundberg & Lee, 2017; Ribeiro et al., 2018). We note that empirical distributions do not necessarily imply feature independence; rather, they can represent complex dependencies extracted from finite datasets (Van den Broeck et al., 2022). The complete formal definitions of these distributions are provided in the Appendix.

## 3 FORMS OF REASONS

In this section, we introduce the explanation types studied in this work, starting with the strict non-probabilistic definitions of (local/global) sufficient and contrastive reasons, and then extending to their more flexible and generalizable probabilistic counterparts.

### 3.1 *Non-Probabilistic* SUFFICIENT AND CONTRASTIVE REASONS

**Sufficient reasons.** In the context of *feature selection*, users often select the top $k$ features that contribute to a model's decision. We examine the well-established *sufficiency* criterion for this selection, which aligns with commonly used explainability methods (Ribeiro et al., 2018; Carter et al., 2019; Ignatiev et al., 2019; Dasgupta et al., 2022). This feature selection can be carried out either locally — focusing on a single prediction — or globally — across the entire input domain. Following standard conventions, we define a *local sufficient reason* as a subset of features $S \subseteq \{1, \dots, n\}$ such that when the features in $S$ are fixed to their corresponding values in $\mathbf{x}$, the model's prediction remains $f(\mathbf{x})$ regardless of the values assigned to the remaining features $\overline{S}$. Formally, $S$ is a local sufficient reason for $\langle f, \mathbf{x} \rangle$ iff the following condition holds:

$$\forall \mathbf{z} \in \mathbb{F}, \quad f(\mathbf{x}_S; \mathbf{z}_{\bar{S}}) = f(\mathbf{x}). \tag{1}$$

Here, $(\mathbf{x}_S; \mathbf{z}_{\bar{S}})$ denotes a vector where features in $S$ take their values from $\mathbf{x}$, and those in $\overline{S}$ from $\mathbf{z}$. Local sufficient reasons are closely related to *semi-factual* explanations (Kenny & Keane, 2021; Alfano et al., 2025), which search for alternative inputs $\mathbf{z}'$ that keep the prediction unchanged (i.e., $f(\mathbf{z}') = f(\mathbf{x})$) while being as "close" as possible to the original point. When we instantiate $\mathbf{z}'$ as $(\mathbf{x}_S; \mathbf{z}_{\bar{S}})$ and measure proximity via Hamming distance, the two notions coincide.

A *global sufficient reason* (Bassan et al., 2024) is a subset of input features $S \subseteq \{1, \dots, n\}$ that serves as a local sufficient reason for *every* possible input $\mathbf{x} \in \mathbb{F}$:

$$\forall \mathbf{x}, \mathbf{z} \in \mathbb{F}, \quad f(\mathbf{x}_S; \mathbf{z}_{\bar{S}}) = f(\mathbf{x}). \tag{2}$$

**Contrastive reasons.** Another prevalent approach to providing explanations is by pinpointing subsets of input features that *modify* a prediction (Dhurandhar et al., 2018; Mothilal et al., 2020; Guidotti, 2024). This type of explanation aims to determine the minimal changes necessary to alter a prediction. Formally, a subset $S \subseteq \{1, \dots, n\}$ is defined as a *local contrastive reason* concerning $\langle f, \mathbf{x} \rangle$ iff:

$$\exists \mathbf{z} \in \mathbb{F}, \quad f(\mathbf{z}_S; \mathbf{x}_{\bar{S}}) \neq f(\mathbf{x}). \tag{3}$$

Contrastive reasons are also closely connected to *counterfactual* explanations (Guidotti, 2024; Mothilal et al., 2020), which seek a nearby assignment $\mathbf{x}'$ for which the prediction flips, i.e., $f(\mathbf{x}') \neq f(\mathbf{x})$. When the distance metric is taken to be the Hamming weight, the two notions coincide.

Similarly to sufficient reasons, one can determine a subset of input features that, when altered, changes the prediction for all inputs within the domain of interest. This form of explanation is also closely connected to approaches for identifying bias (Arenas et al., 2021; Bassan et al., 2024; Darwiche & Hirth, 2020), as well as to *group* counterfactual explanation methods that search for counterfactuals over multiple data instances (Carrizosa et al., 2024; Warren et al., 2024). Formally, we define a subset $S$ as a *global contrastive reason* with respect to $f$ iff:

$$\forall \mathbf{x} \in \mathbb{F}, \quad \exists \mathbf{z} \in \mathbb{F}, \quad f(\mathbf{z}_S; \mathbf{x}_{\bar{S}}) \neq f(\mathbf{x}). \tag{4}$$

### 3.2 *Probabilistic* SUFFICIENT AND CONTRASTIVE REASONS

As discussed in the introduction, non-probabilistic sufficient and contrastive reasons are often criticized for imposing significantly overly strict conditions, motivating a shift to *probabilistic* definitions (Arenas et al., 2022; Ribeiro et al., 2018; Izza et al., 2023; Wäldchen et al., 2021; Bounia & Koriche, 2023; Subercaseaux et al., 2025; Wang et al., 2021a; Blanc et al., 2021), which generalize these requirements by demanding that the guarantees hold with probability at least $\delta \in [0, 1]$. The special case $\delta = 1$ recovers the original non-probabilistic definitions.

**Probabilistic sufficient reasons.** We define $S \subseteq \{1, \dots, n\}$ as a *local $\delta$-sufficient reason* with respect to $\langle f, \mathbf{x} \rangle$ if, when the features in $S$ are fixed to their corresponding values in $\mathbf{x}$ and the remaining features are sampled from a distribution $\mathcal{D}$, the classification remains unchanged with probability at least $\delta$. In other words:

$$\mathbf{Pr}_{\mathbf{z} \sim \mathcal{D}}(f(\mathbf{z}) = f(\mathbf{x}) \mid \mathbf{z}_S = \mathbf{x}_S) \geq \delta. \tag{5}$$

where $\mathbf{z}_S = \mathbf{x}_S$ denotes that the features in $S$ of vector $\mathbf{z}$ are fixed to their corresponding values in $\mathbf{x}$. We adopt the standard notion of global explanations — by averaging over all inputs in the global domain — and define a subset $S \subseteq \{1, \dots, n\}$ as a *global $\delta$-sufficient reason* with respect to $\langle f \rangle$ if, when taking the expectation of the local sufficiency probability over samples drawn from the distribution $\mathcal{D}$, the expectation remains with value at least $\delta$. In other words:

$$\mathbb{E}_{\mathbf{x} \sim \mathcal{D}}[\mathbf{Pr}_{\mathbf{z} \sim \mathcal{D}}(f(\mathbf{z}) = f(\mathbf{x}) \mid \mathbf{z}_S = \mathbf{x}_S)] \geq \delta. \tag{6}$$

**Probabilistic contrastive reasons.** Similar to the non-probabilistic case, we define a *local $\delta$-contrastive reason* as a subset of input features that changes a prediction with some probability. Here, unlike sufficient reasons, we set the features of the *complementary* set $\overline{S}$ to their respective values in $\mathbf{x}$, and when allowing the features in $S$ to vary, we require the prediction to differ from the original prediction $f(\mathbf{x})$ with a probability exceeding $\delta$. Formally:

$$\mathbf{Pr}_{\mathbf{z} \sim \mathcal{D}}(f(\mathbf{z}) \neq f(\mathbf{x}) \mid \mathbf{z}_{\overline{S}} = \mathbf{x}_{\overline{S}}) \geq \delta. \tag{7}$$

For the global setting, we define a subset $S$ to be a *global $\delta$-contrastive reason*, analogous to global sufficient reasons, by computing the expectation over all local contrastive reasons sampled from the distribution $\mathcal{D}$, and requiring that this expected value exceeds $\delta$:

$$\mathbb{E}_{\mathbf{x} \sim \mathcal{D}}[\mathbf{Pr}_{\mathbf{z} \sim \mathcal{D}}(f(\mathbf{z}) \neq f(\mathbf{x}) \mid \mathbf{z}_{\overline{S}} = \mathbf{x}_{\overline{S}})] \geq \delta. \tag{8}$$

## 4 FORMS OF MINIMALITY

As discussed in the introduction, across all the explanation forms discussed so far — whether sufficient or contrastive, local or global — a common assumption in the literature is that explanations of *smaller size* are more meaningful, thereby making their *minimality* a particularly important provable guarantee. In this study, we explore two central notions of minimality across all our explanation types:

**Definition 1.** *Assuming a subset $S \subseteq \{1, \dots n\}$ is an explanation, then:*

1. *$S$ is a* cardinally-minimal *explanation (Barceló et al., 2020; Bassan et al., 2024) iff $S$ has the smallest explanation cardinality $|S|$ (i.e., there is no explanation $S'$ such that $|S'| < |S|$).*

2. *$S \subseteq \{1, \dots, n\}$ is a* subset-minimal *explanation (Arenas et al., 2022; Ignatiev et al., 2019) iff $S$ is an explanation, and any $S' \subseteq S$ is not an explanation.*

We note that cardinal-minimality is strictly stronger than subset-minimality: every cardinally minimal $S$ is subset-minimal, but not vice versa (see Appendix B.3 for an example). We use the terms subset and cardinally minimal, rather than local and global minima, to avoid confusion with local vs. global explanations (input- vs. domain-level reasoning). Both notions apply to *all* explanation forms we analyze. For instance, a cardinally minimal local probabilistic $\delta$ sufficient reason is the one with the smallest $|S|$, while a subset-minimal one is any $S$ where no proper subset $S' \subseteq S$ also qualifies.

## 5 A Unified Combinatorial Optimization Task

Interestingly, all previously discussed computational problems — local or global, sufficient or contrastive, probabilistic or not — can be cast as finding a minimal-size subset $S$ such that $v(S) \geq \delta$, where in non-probabilistic settings we set $\delta := 1$. We now introduce notation for the value functions: let $v_{\text{suff}}^{\ell}$ denote the local sufficiency probability from equation 5, i.e., $\mathbf{Pr}_{\mathbf{z} \sim \mathcal{D}_p}(f(\mathbf{x}) = f(\mathbf{z}) \mid \mathbf{x}_S = \mathbf{z}_S)$, and define the global variant as $v_{\text{suff}}^{g}$ (equation 6). Similarly, let $v_{\text{con}}^{\ell}$ and $v_{\text{con}}^{g}$ denote the local and global contrastive probabilities (equations 7 and 73, respectively). Using this notation, we now formally define the unified task of finding a cardinally minimal $\delta$-local/global sufficient/contrastive reason:

---

**Cardinally Minimal $\delta$-Explanation**:
**Input**: Model $f$, a distribution $\mathcal{D}$, (possibly, an input $\mathbf{x}$), a *general* value function $v : 2^n \to [0, 1]$ (defined using $f$, $\mathcal{D}$, and possibly $\mathbf{x}$), and some $\delta \in [0, 1]$.
**Output**: A subset $S \subseteq [n]$ such that $v(S) \geq \delta$ and $|S|$ is minimal.

---

Similarly, for the relaxed condition where the goal is to obtain a subset-minimal rather than a cardinally-minimal local or global sufficient/contrastive reason, we define the following relaxed optimization objective:

---

**Subset Minimal $\delta$-Explanation**:
**Input**: Model $f$, a distribution $\mathcal{D}$, (possibly, an input $\mathbf{x}$), a *general* value function $v : 2^n \to [0, 1]$ (defined using $f$, $\mathcal{D}$, and possibly $\mathbf{x}$), and some $\delta \in [0, 1]$.
**Output**: A subset $S \subseteq [n]$ such that $v(S) \geq \delta$ and for any $S' \subseteq S$ it holds that $v(S') < \delta$.

---

**Properties of $v$ that affect the complexity.** We now outline several key properties of the value function, which we later show play a crucial role in determining the complexity of generating explanations. The first property is *non-decreasing monotonicity*, which ensures that the marginal contribution $v(S \cup \{i\}) - v(S)$ is consistently non-negative. Formally:

**Definition 2.** *We say that a value function $v$ maintains* non-decreasing monotonicity *if for any $S \subseteq \{1, \ldots, n\}$ and any $i \in \{1, \ldots, n\}$ it holds that: $v(S \cup \{i\}) \geq v(S)$.*

The other key properties of *supermodularity* and *submodularity* pertain to the behavior of the marginal contribution $v(S \cup \{i\}) - v(S)$. Specifically, in the supermodular case, this contribution forms a monotone non-decreasing function. In contrast, under the dual definition of the *submodular* case, the marginal contribution $v(S \cup \{i\}) - v(S)$ is a monotone non-increasing function. Formally:

**Definition 3.** *Let there be some value function $v$, some $S \subseteq S' \subseteq \{1, \ldots, n\}$, and $i \notin S'$. Then:*

1. *$v$ maintains* supermodularity *iff it holds that: $v(S \cup \{i\}) - v(S) \leq v(S' \cup \{i\}) - v(S')$.*

2. *$v$ maintains* submodularity *iff it holds that: $v(S \cup \{i\}) - v(S) \geq v(S' \cup \{i\}) - v(S')$.*

## 6 Unraveling New Properties of the Global Value Functions

Prior work shows that in the local setting of non-probabilistic explanations, subset-minimal sufficient or contrastive reasons can be computed thanks to monotonicity, which holds only in the restricted case $\delta = 1$. This assumption, however, is highly limiting and lacks practical flexibility. While one might hope to generalize to probabilistic guarantees for arbitrary $\delta$, prior results demonstrate that monotonicity breaks down in this setting, rendering the computation of explanations computationally harder (Arenas et al., 2022; Kozachinskiy, 2023; Izza et al., 2023; Subercaseaux et al., 2025; Izza et al., 2024b).

At the even more extreme case of the *global* and *non-probabilistic* setting, Bassan et al. (2024) demonstrated the stringent *uniqueness* property — i.e., there is exactly *one* subset-minimal explanation. However, requiring $\delta = 1$ makes this setting highly restrictive, especially in the global case, where the explanation conditions must hold for *all* inputs. In fact, Bassan et al. (2024) proves that this unique minimal subset is equivalent to the subset of all features that are not strictly redundant (which may, in practice, be all of them). We prove that in the general global *probabilistic* setting — for any $\delta$ — this uniqueness property actually does *not* hold, and the number of subsets can even be *exponential*.

**Proposition 1.** *While the non-probabilistic case ($\delta = 1$) admits a unique subset-minimal global sufficient reason (Bassan et al., 2024), in the general probabilistic setting (for arbitrary $\delta$), there exist functions $f$ that have $\Theta(\frac{2^n}{\sqrt{n}})$ subset-minimal global sufficient reasons.*

Interestingly, although the uniqueness property fails to hold in the general case for arbitrary $\delta$ (and is restricted to the special case of $\delta = 1$), we show that the crucial *monotonicity* property holds for the global value function across all values of $\delta$. This applies to both the sufficient ($v_{\text{suff}}^g$) and contrastive ($v_{\text{con}}^g$) value functions. This finding is surprising as it stands in sharp contrast to the local setting, where the corresponding value functions ($v_{\text{suff}}^\ell$ and $v_{\text{con}}^\ell$) do not satisfy this property:

**Proposition 2.** *While the local probabilistic setting (for any $\delta$) lacks monotonicity — i.e., the value functions $v_{con}^\ell$ and $v_{suff}^\ell$ are non-monotonic (Arenas et al., 2022; Izza et al., 2023; Subercaseaux et al., 2025; Izza et al., 2024b) — in the global probabilistic setting (also for any $\delta$), both value functions $v_{con}^g$ and $v_{suff}^g$ are monotonic non-decreasing.*

Beyond the surprising insight that monotonicity holds for the global value functions — but fails to hold in the local one — we identify additional structural properties unique to the global setting, including submodularity or supermodularity. In particular, we show that under the common assumption of feature independence, the global sufficient value function $v_{\text{suff}}^g$ exhibits supermodularity. In contrast, its local counterpart $v_{\text{suff}}^\ell$ fails to exhibit this property even under the much more restrictive assumption of a uniform (and hence independent) input distribution. Specifically:

**Proposition 3.** *While the local probabilistic sufficient setting (for any $\delta$) lacks supermodularity — even when $\mathcal{D}$ is uniform, i.e., the value function $v_{suff}^\ell$ is not supermodular — in the global probabilistic setting (also for any $\delta$), when $\mathcal{D}$ exhibits feature independence, the value function $v_{suff}^g$ is supermodular.*

Interestingly, for the second family of explanation settings — specifically, that of obtaining a global probabilistic *contrastive* reason — we show that the corresponding value function is not supermodular, but rather *submodular*. This result is particularly surprising when contrasted with the local setting, where the value function is neither submodular nor supermodular.

**Proposition 4.** *While the local probabilistic contrastive setting (for any $\delta$) lacks submodularity — even when $\mathcal{D}$ is uniform, i.e., the value function $v_{con}^\ell$ is not submodular — in the global probabilistic setting (also for any $\delta$), when $\mathcal{D}$ exhibits feature independence, the value function $v_{con}^g$ is submodular.*

## 7 COMPUTATIONAL COMPLEXITY RESULTS

### 7.1 SUBSET MINIMAL EXPLANATIONS

In this section, we examine the complexity of obtaining *subset minimal* explanations (local/global, sufficient/contrastive) across the different model types analyzed. The key property at play here is the *monotonicity* of the value function. The previous section established that monotonicity holds for both global value functions, $v_{\text{con}}^g$ and $v_{\text{suff}}^g$, but does not hold for the local value functions, $v_{\text{con}}^\ell$ and $v_{\text{suff}}^\ell$. This distinction is crucial in showing that a greedy algorithm converges to a subset minimal explanation in the global setting but fails in the local setting. As a result, we will showcase the surprising finding that computing various local subset-minimal explanation forms is hard, whereas computing subset-minimal global explanation forms is tractable (polynomial-time solvable). We will begin by introducing the following generalized greedy algorithm:

---
**Algorithm 1** Subset Minimal Explanation Search
---
**Input** Value function $v$, and some $\delta \in [0, 1]$
 1: $S \leftarrow \{1, \ldots, n\}$
 2: **while** $\min_{i \in S} v(S \setminus \{i\}) \geq \delta$ **do**
 3:     $j \leftarrow \operatorname{argmax}_{i \in S} v(S \setminus \{i\})$
 4:     $S \leftarrow S \setminus \{j\}$
 5: **end while**
 6: **return** $S$                  $\triangleright$ $S$ is a (subset minimal?) $\delta$-explanation

---

Algorithm 1 aims to obtain a subset-minimal $\delta$-explanation. We begin the algorithm with the subset $S$ initialized as the entire input space $\{1, \ldots, n\}$. Iteratively, we check whether the minimal value that

$v(S \setminus \{i\})$ can attain exceeds $\delta$. In each iteration, we remove a feature $j$ from $S$ that minimizes the decrease in the value function, selecting the feature $j$ that maximizes $v(S \setminus \{j\})$. Once this iterative process concludes, we return $S$.

The key determinant of whether Algorithm 1 yields a subset-minimal explanation is the *monotonicity* property of the value function $v$. The algorithm concludes with a phase in which removing any individual feature from $S$ results in $v(S \setminus \{i\})$ being smaller than $\delta$. However, monotonicity ensures that this holds for any $v(S \setminus S')$, providing a significantly stronger guarantee. Given the monotonicity property of the value functions established in the previous sections, we derive the following proposition:

**Proposition 5.** *Computing Algorithm 1 with the* local *value functions $v_{con}^{\ell}$ and $v_{suff}^{\ell}$ does not always converge to a subset minimal $\delta$-sufficient/contrastive reason. However, computing it with the* global *value functions $v_{con}^{g}$ or $v_{suff}^{g}$ necessarily produces subset minimal $\delta$-sufficient/contrastive reasons.*

Building on this result, we proceed to establish new complexity findings for deriving various forms of subset-minimal explanations within our framework, considering the different analyzed model types.

**Decision trees.** We begin by examining a highly simplified and ostensibly "interpretable" scenario. Specifically, we assume that the model $f$ is a decision tree and that the distribution $\mathcal{D}$ is independent. Within this simplified setting, we demonstrate a strict separation: Arenas et al. (2022) established the surprising intractability result that, unless PTIME=NP, no polynomial-time algorithm exists for computing a subset-minimal local $\delta$-sufficient reason for decision trees under independent distributions (even under the uniform distribution). In contrast, we demonstrate the unexpected result that this exact problem can be solved efficiently in the global setting, meaning that a subset-minimal global $\delta$-sufficient reason for decision trees can indeed be computed in polynomial time. Formally:

**Theorem 1.** *If $f$ is a decision tree and the probability term $v_{suff}^{g}$ can be computed in polynomial time given the distribution $\mathcal{D}$ (which holds for independent distributions, among others), then obtaining a subset-minimal* global *$\delta$-sufficient reason can be obtained in* polynomial time. *However, unless PTIME=NP,* no polynomial-time *algorithm exists for computing a* local *$\delta$-sufficient reason for decision trees even under independent distributions.*

**Extension to other tractable models.** We additionally note that a similar tractability guarantee for computing global explanations also holds for *orthogonal DNFs* (Crama & Hammer, 2011), which we briefly recall are Disjunctive Normal Form formulas whose terms are pairwise mutually exclusive. Establishing the result for this class is useful because orthogonal DNFs *generalize* decision trees while preserving their clean structural properties, allowing our guarantees to extend beyond tree-structured models. For completeness, we provide the full argument in Appendix N.

**Neural networks, tree ensembles, and other complex models.** We now extend our previous results to more complex models beyond decision trees. Specifically, we will demonstrate that when the distribution $\mathcal{D}$ is derived from empirical distributions, and under the fundamental assumption that the model $f$ allows polynomial-time inference, it follows that computing a subset-minimal global $\delta$-sufficient and contrastive reason can be done in polynomial time.

**Proposition 6.** *For any model $f$, and empirical distribution $\mathcal{D}$ — computing a subset-minimal* global *$\delta$-sufficient or $\delta$-contrastive reason for $f$ can be done in polynomial time.*

This strong complexity outcome, which holds for *any* model under an empirical data distribution assumption, allows us to further differentiate the complexity of local and global explanation settings. Specifically, for certain complex models, computing subset-minimal *local* explanations remains intractable even when restricted to empirical distributions. We establish this fact for both neural networks and tree ensembles, leading to the following theorem on a strict complexity separation:

**Theorem 2.** *Assuming $f$ is a neural network or a tree ensemble, and $\mathcal{D}$ is an empirical distribution — there exist* polynomial-time *algorithms for obtaining subset minimal* global *$\delta$-sufficient and contrastive reasons. However, unless PTIME=NP, there is* no polynomial time *algorithm for computing a subset minimal* local *$\delta$-sufficient reason or a subset minimal* local *$\delta$-contrastive reason.*

## 7.2 Approximate Cardinally Minimal Explanations

In this subsection, we shift our focus from subset-minimal sufficient/contrastive reasons to the even more challenging task of finding a *cardinally minimal $\delta$-sufficient/contrastive reason*. We

will demonstrate that in the global setting, the interplay between supermodularity/submodularity and monotonicity of the value function enables us to establish novel *provable approximations* for computing explanations. In contrast, we will show that computing these explanations in the local setting remains intractable. This result further strengthens the surprising distinction between the tractability of computing global explanations versus local ones.

**A unified greedy approximation algorithm.** When working with a non-decreasing monotonic submodular function, the problem of identifying a cardinally minimal explanation closely aligns with the *submodular set cover* problem (Wolsey, 1982). In contrast, employing a *supermodular* function leads to a non-submodular variation of this problem (Shi et al., 2021). These problems have garnered significant interest due to their strong approximation guarantees (Wolsey, 1982; Iyer & Bilmes, 2013; Chen & Crawford, 2023). In the context of submodular set cover optimization, a standard approach involves using a classic greedy algorithm, which we will first outline (Algorithm 2). This algorithm serves as the foundation for approximating a cardinally minimal sufficient or contrastive $\delta$-reason, and we will later examine its specific guarantees in both cases.

---

**Algorithm 2** Cardinally Minimal Explanation Approximation Search

---

**Input** Value function $v$, and some $\delta \in [0, 1]$

1:   $S \leftarrow \emptyset$
2: **while** $\max_{i \in S} v(S \cup \{i\}) < \delta$ **do**
3:     $j \leftarrow \arg\max_{i \in S} v(S \cup \{i\})$
4:     $S \leftarrow S \cup \{j\}$
5: **end while**
6: **return** $S$         ▷ $|S|$ is a (provable approximation?) of a cardinally minimal $\delta$-explanation

---

Algorithm 2 closely resembles Algorithm 1, but works bottom-up rather than top-down. It starts with an empty subset and incrementally adds features, each time selecting the one that minimizes the increase in $v(S \cup i)$, stopping when adding any feature would push the value over $\delta$.

**Cardinally minimal contrastive reasons.** We begin with global contrastive reasons, where monotonicity and submodularity hold, reducing the task to the classic *submodular set cover* problem and allowing us to apply Wolsey (1982)'s classic guarantee via Algorithm 2. For integer-valued objectives, the algorithm achieves a Harmonic-based factor, and more generally an $\mathcal{O}\left(\ln\left(\frac{v([n])}{\min_{i \in [n]} v(i)}\right)\right)$-approximation. Under an *empirical* distribution $\mathcal{D}$ with fixed sample size, this becomes a *constant approximation*, only *logarithmic* in the sample size, yielding a substantially strong guarantee. By contrast, in the *local* setting — even for a *single* sample point — no *bounded* approximation exists, marking a sharp gap between global and local cases.

**Theorem 3.** *Given a neural network or tree ensemble $f$ and an empirical distribution $\mathcal{D}$ over a fixed dataset $D$, Algorithm 2 yields a* constant $\mathcal{O}\left(\ln\left(\frac{v_{con}^g([n])}{\min_{i \in [n]} v_{con}^g(\{i\})}\right)\right)$-*approximation, bounded by* $\mathcal{O}(\ln(|D|))$, *for computing a global cardinally minimal $\delta$-contrastive reason for $f$, assuming feature independence. In contrast, unless PTIME=NP, no bounded approximation exists for computing a local cardinally minimal $\delta$-contrastive reason for any $\langle f, \boldsymbol{x} \rangle$, even when $|D| = 1$.*

We also provide a matching *lower bound* for the bound in Theorem 3 in the specific case of an empirical distribution without any independence assumption (see Appendix M).

**Cardinally minimal sufficient reasons.** Unlike the submodular set cover problem — linked to cardinally minimal global *contrastive* reasons and admitting strong approximations — the supermodular variant, tied to global *sufficient* reasons, is harder to approximate. Still, it offers guarantees when the function has bounded curvature (Shi et al., 2021). The *total curvature* of a function $v : 2^n \to \mathbb{R}$ is defined as $k^f := 1 - \min_{i \in [n]} \frac{v([n]) - v([n] \setminus i)}{v(i) - v(\emptyset)}$. Leveraging results from Shi et al. (2021), we show that Algorithm 2 achieves an $\mathcal{O}\left(\frac{1}{1-k^f} + \ln\left(\frac{v([n])}{\min_{i \in [n]} v(i)}\right)\right)$-approximation.

Notably, under a fixed empirical distribution, the approximation becomes constant. While contrastive reasons admit a tighter $\mathcal{O}(\ln(|D|))$ bound, sufficient reasons incur an extra $\frac{1}{1-k^f}$ factor — yet still yield a constant-factor approximation. In sharp contrast, the local variant remains inapproximable, lacking any *bounded* approximation even when $|D| = 1$.

**Theorem 4.** *Given a neural network or tree ensemble $f$ and an empirical distribution $\mathcal{D}$ over a fixed dataset $D$, Algorithm 2 yields a* constant $\mathcal{O}\left(\frac{1}{1-k^f} + \ln\left(\frac{v_{suff}^g([n])}{\min_{i \in [n]} v_{suff}^g(\{i\})}\right)\right)$-*approximation for computing a global cardinally minimal $\delta$-sufficient reason for $f$, assuming feature independence. In contrast, unless PTIME=NP, there is* no bounded approximation *for computing a local cardinally minimal $\delta$-sufficient reason for any $\langle f, \boldsymbol{x} \rangle$, even when $|D| = 1$.*

Overall, these findings strengthen Subsection 7.1, which showed the tractability of computing subset-minimal global explanations in stark contrast to local ones. Here, we further show that approximating cardinally minimal global explanations is tractable, unlike their inapproximable local counterparts.

## 8 LIMITATIONS AND FUTURE WORK

While many of our most important findings — particularly those concerning fundamental properties of value functions — hold generally, we also instantiate them to yield concrete complexity results for specific model classes (e.g., neural networks), distributional assumptions (e.g., empirical distributions), and explanation definitions within our framework. Naturally, other potential settings remain open for analysis. Nonetheless, we believe our findings offer compelling insights into foundational aspects of explanations, along with new tractability and intractability results, which together lay a strong foundation for investigating a broader range of explainability scenarios in future work.

More specifically, our study brings forward several important open theoretical questions:

1. Firstly, it would be interesting to investigate whether some of the tractable complexity results we obtained for global explainability can be tightened for simpler model classes, such as decision trees, linear models, or other models with inherently tractable structure. Although local explanations for such models are often intractable to compute (Arenas et al., 2022; Subercaseaux et al., 2025), the global explanation forms we study in this work exhibit strong structural properties, enabling significantly improved complexity guarantees that may be tightened even further. Furthermore, tightening the approximation guarantees when working with empirical distributions represents a key open direction.

2. Secondly, when *exact* computation of expectations is infeasible, it is natural to ask whether alternative techniques — such as Fully Polynomial Randomized Approximation Schemes (FPRAS) for model classes like DNF formulas (Meel et al., 2019) (and thus for tree ensembles), or Monte Carlo–based approximations as the ones used in (Subercaseaux et al., 2025) — can be employed to approximate these expectations. A key open challenge is determining how to maintain minimality guarantees under approximation error, and what tolerance levels still allow the guarantees to hold.

3. Finally, we believe that studying alternative notions of feature importance — e.g., other loss measures such as KL divergence (Conmy et al., 2023) — may reveal similar patterns of monotonicity, submodularity, or supermodularity when transitioning from local to global explanations. Such properties could enable efficient feature selection under these alternative metrics as well.

## 9 CONCLUSION

We present a unified framework for evaluating diverse explanations and reveal a stark contrast between local and global sufficient and contrastive reasons. Notably, while the *local* explanation variants lack any structural form common in combinatorial optimization — such as monotonicity, submodularity, or supermodularity, we prove that their *global* counterparts exhibit precisely these crucial properties: (i) *monotonicity*, (ii) *submodularity* in the case of contrastive reasons, and (iii) *supermodularity* for sufficient reasons. These proofs form the basis for proving a series of surprising complexity results, showing that global explanations with provable guarantees can be computed efficiently, even for complex model classes such as neural networks. In sharp contrast, we prove that computing the corresponding local explanations remains NP-hard — even in highly simplified scenarios. Altogether, our results uncover foundational properties of explanations and chart both tractable and intractable frontiers, opening new avenues for future research.

## ACKNOWLEDGMENTS

This work was partially funded by the European Union (ERC, VeriDeL, 101112713). Views and opinions expressed are however those of the author(s) only and do not necessarily reflect those of the European Union or the European Research Council Executive Agency. Neither the European Union nor the granting authority can be held responsible for them. This research was additionally supported by a grant from the Israeli Science Foundation (grant number 558/24).

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

# Appendix

The appendix contains formalizations and proofs that were mentioned throughout the paper:

**Appendix A** contains extended related work on Formal XAI.
**Appendix B** contains the formalizations of the models and distributions used in this work.
**Appendix C** contains the proof of Proposition 1.
**Appendix D** contains the proof of Proposition 2.
**Appendix E** contains the proof of Proposition 3.
**Appendix F** contains the proof of Proposition 4.
**Appendix G** contains the proof of Proposition 5.
**Appendix H** contains the proof of Theorem 1.
**Appendix I** contains the proof of Proposition 6.
**Appendix J** contains the proof of Theorem 2.
**Appendix K** contains the proof of Theorem 3.
**Appendix L** contains the proof of Theorem 4.
**Appendix O** contains an LLM usage disclosure.

## A  EXTENDED RELATED WORK ON FORMAL XAI

Our work falls within the line of research on *formal explainable AI* (formal XAI) (Marques-Silva & Ignatiev, 2022), which studies explanations equipped with provable guarantees (Yu et al., 2023; Darwiche & Ji, 2022; Darwiche, 2023; Shih et al., 2018; Azzolin et al., 2025; Audemard et al., 2021; Calautti et al., 2025). A central theme in this area concerns the computational complexity of generating such guaranteed explanations (Barceló et al., 2020; **?**; Marzouk et al., 2025a;b; Marzouk & De La Higuera, 2024; Bassan et al., 2025e; Amir et al., 2024; Blanc et al., 2021; 2022; Jaakkola et al., 2025; Amgoud & Cooper, 2026; Geibinger et al., 2025). Since obtaining formal explanations is often computationally intractable for expressive models such as neural networks (Adolfi et al., 2024; Barceló et al., 2020) and tree ensembles (Ordyniak et al., 2024; Bassan et al., 2025a), much of the literature has focused on more tractable model classes (Marques-Silva & Ignatiev, 2023), including decision trees (Bounia, 2025; Arenas et al., 2022; Bounia, 2024; Bounia & Koriche, 2023), monotonic models (Marques-Silva et al., 2021; Harzli et al., 2023), linear models (Marques-Silva et al., 2020; Subercaseaux et al., 2025), and additive models (Bassan et al., 2026).

Beyond complexity-theoretic investigations, substantial effort has also been devoted to computing formal explanations in practice using automated reasoning tools, such as MaxSAT and MILP solvers for tree ensembles (Audemard et al., 2023; 2022a; Ignatiev et al., 2022; Boumazouza et al., 2021), and neural network verification frameworks (Wang et al., 2021b; Wu et al., 2024a) to derive certified explanations for neural networks (Wu et al., 2024b; Bassan & Katz, 2023; Bassan et al., 2023; Izza et al., 2024a; Hadad et al., 2026; Ignatiev et al., 2019). Additional approaches seek to alleviate the inherent computational burden through abstractions (De Palma et al., 2025; Bassan et al., 2025b; Boumazouza et al., 2026), relaxations such as smoothing (Xue et al., 2023; Anani et al., 2025; Jin et al., 2025), or training-time interventions (Alvarez Melis & Jaakkola, 2018; Bassan et al., 2025c;d).

Finally, within formal XAI, a related line of work studies methods that leverage combinatorial and submodular optimization techniques to improve the computation of *local* probabilistic sufficient reasons (Bounia, 2025; Bounia & Koriche, 2023), particularly for decision tree classifiers. As noted in that literature, as well as in this work, the value-function objective underlying these local explanation variants is not submodular in practice. Nevertheless, greedy algorithms inspired by submodular optimization often serve as effective heuristics and can yield high-quality explanations empirically. In contrast, our work establishes a rigorous theoretical connection between combinatorial and submodular optimization objectives and the *global* variants of sufficient and contrastive reasons. This connection provides formal guarantees and opens the door to a broader range of principled algorithmic implementations.

# B    MODEL AND DISTRIBUTION FORMALIZATIONS

In this section, we formalize the models and distributions referenced throughout the paper. Specifically, Subsection B.1 defines the model families, while Subsection B.2 formalizes the distributions.

## B.1    MODEL FORMALIZATIONS

In this subsection, we formalize the three base-model types that were analyzed throughout the paper: (i) (axis-aligned) decision trees, (ii) linear classifiers, and (iii) neural networks with ReLU activations.

**Decision Trees.** We define a decision tree (DT) as a directed acyclic graph that represents a function $f : \mathbb{F} \to [c]$, where $c \in \mathbb{N}$ denotes the number of classes. The graph encodes the function as follows: (i) Each internal node $v$ is assigned a distinct binary input feature from the set $\{1, \ldots, n\}$; (ii) Every internal node $v$ has at most $k$ outgoing edges, each corresponding to a value in $[k]$ assigned to $v$; (iii) Along any path $\alpha$ in the decision tree, each variable appears at most once; (iv) Each leaf node is labeled with a class from $[c]$. Thus, assigning values to the inputs $\mathbf{x} \in \mathbb{F}$ uniquely determines a path $\alpha$ from the root to a leaf in the DT, where the function output $f(\mathbf{x})$ corresponds to a class label $i \in [c]$. The size of the DT, denoted $|f|$, is defined as the total number of edges in the graph. To allow for flexible modeling, the ordering of input variables $\{1, \ldots, n\}$ may differ across distinct paths $\alpha$ and $\alpha'$, ensuring that no variable is repeated along a single path.

**Neural Networks.** We present our *hardness* proofs for neural networks with ReLU activations, though our *tractability results* (i.e., polynomial-time algorithms) apply to any architecture that allows for polynomial-time inference — a standard assumption. Thus, all separation results between tractable and intractable cases we prove carry over to *any neural architecture* at least as expressive as a standard feed-forward ReLU network, encompassing many widely used models such as ResNets, CNNs, Transformers, Diffusion models, and more. Moreover, note that any ReLU network can be represented as a fully connected network by assigning zero weights and biases to missing connections. Following standard conventions (Barceló et al., 2020; Bassan et al., 2024; Adolfi et al., 2025), we thus assume the network is fully connected. Specifically, our analysis applies to multi-layer perceptrons (MLPs). Formally, an MLP $f$ consists of $t - 1$ *hidden layers* ($g^{(j)}$ for $j = 1, \ldots, t - 1$) and one output layer $g^{(t)}$, where each layer is defined recursively as:

$$g^{(j)} := \sigma^{(j)}(g^{(j-1)} W^{(j)} + b^{(j)}) \tag{9}$$

where $W^{(j)}$ denotes the weight matrix, $b^{(j)}$ the bias vector, and $\sigma^{(j)}$ the activation function of the $j$-th layer. Accordingly, the model comprises $t$ weight matrices $(W^{(1)}, \ldots, W^{(t)})$, $t$ bias vectors $(b^{(1)}, \ldots, b^{(t)})$, and $t$ activation functions $(\sigma^{(1)}, \ldots, \sigma^{(t)})$.

The input layer is defined as $g^{(0)} = \mathbf{x} \in \{0, 1\}^n$, representing the binary input vector. The dimensions of the network are governed by a sequence of positive integers $d_0, \ldots, d_t$, with weight matrices and bias vectors given by $W^{(j)} \in \mathbb{Q}^{d_{j-1} \times d_j}$ and $b^{(j)} \in \mathbb{Q}^{d_j}$, respectively. These parameters are learned during training. Since the model functions as a binary classifier over $n$ features, we set $d_0 = n$ and $d_t = 1$. The hidden layers use the ReLU activation function, defined by $\mathrm{ReLU}(x) = \max(0, x)$. Although a sigmoid activation is typically used during training, for interpretability purposes, we assume the output layer applies a threshold-based step function, defined as $\mathrm{step}(\mathbf{z}) = 1$ if $\mathbf{z} \geq 0$ and $\mathrm{step}(\mathbf{z}) = 0$ otherwise.

**Tree Ensembles.** Many popular ensemble methods exist, but since our goal is to provide post-hoc explanations, we focus on the *inference* phase rather than the training process. Our analysis centers on ensemble families that rely on either *majority voting* (e.g., Random Forests) or *weighted voting* (e.g., XGBoost) during inference. However, as with our treatment of neural networks, our *tractability* results — namely, polynomial-time algorithms — extend to any possible ensemble configuration with polynomial-time inference, encompassing an even broader range of ensemble configurations.

**Majority Voting Inference.** In *majority voting* inference, the final prediction $f(\mathbf{x})$ is assigned to the class $j \in [c]$ that receives the majority of votes among the individual predictions $f_i(\mathbf{x})$ from all $i \in [k]$ (i.e., from each tree in the ensemble).

$$f(\mathbf{x}) := \text{majority}(\{f_i(\mathbf{x}) \mid i \in [k]\}) \tag{10}$$

where $\text{majority}(S)$ denotes the most frequent label in the multiset $S$. If there is a tie, it is resolved by a fixed tie-breaking rule (e.g., lexicographic order or predefined priority).

**Weighted Voting Inference.** In *weighted voting* inference, each model in the ensemble is assigned a weight $\phi_i \in \mathbb{Q}$ representing its relative importance. The predicted class is the one with the highest total weight across all models. Formally, for any $\mathbf{x} \in \mathbb{F}$, we define $f$ as:

$$f(\mathbf{x}) := \arg\max_{j \in [c]} \sum_{i=1}^{k} \phi_i \cdot \mathbb{I}[f_i(\mathbf{x}) = j] \tag{11}$$

where $\mathbb{I}$ denotes the identity function.

## B.2 DISTRIBUTION FORMALIZATIONS

This subsection formalizes the distribution definitions discussed in the main paper.

**Empirical Distributions.** The distribution $\mathcal{D}$ over the input features will be defined based on a dataset $\mathbf{D}$ comprising various inputs $\mathbf{z}^1, \mathbf{z}^2, \ldots, \mathbf{z}^{|\mathbf{D}|}$. Here, the distribution of any given input $\mathbf{x} \in \mathbb{F}$ is defined by the frequency of occurrence of $\mathbf{x}$ within $\mathbf{D}$, specifically by:

$$\mathbf{Pr}(\mathbf{x}) := \frac{1}{|\mathbf{D}|} \sum_{i=1}^{|\mathbf{D}|} \mathbb{I}(\mathbf{z}^i = \mathbf{x}) \tag{12}$$

**Independent Distributions.** Formally, given a probability value $p(\mathbf{x}_i) \in [0, 1]$ defined for each individual input feature, we say that the distribution $\mathcal{D}$ is independent if the joint probability over inputs is given by $\mathbf{Pr}(\mathbf{x}) := \prod_{i \in [n]} p(\mathbf{x}_i)$.

We observe that when $p(\mathbf{x}_i) = p(\mathbf{x}_j)$ holds for all $i, j \in [n]$, the distribution reduces to the *uniform distribution*, a special case of independent distributions.

**General Distributions.** We note that many of the proofs in this work — particularly those concerning fundamental properties of value functions — apply broadly over general distribution assumptions. While empirical distributions (using the training dataset as a proxy) are common in XAI, alternative frameworks for approximating distributions include k-NN resampling from nearby points (Li et al., 2023; Almanjahie et al., 2018; Gweon et al., 2019), copulas for modeling tabular dependencies (Größer & Okhrin, 2022), or more advanced conditional generative models such as CTGAN (Xu et al., 2019) and conditional diffusion models (Huang et al., 2022). Such choices are common in the counterfactual explanation and algorithmic recourse literature (Karimi et al., 2021; Fokkema et al., 2024; 2023; Verma et al., 2024), which are related to contrastive explanations. Furthermore, when the value function is not computed directly from a structural property of the model (e.g., leaf enumeration in decision trees) or from empirical distributions, one may instead approximate it using methods such as Monte Carlo sampling (Hastings, 1970), following approaches similar to (Subercaseaux et al., 2025; Lopardo et al., 2023).

## B.3 SUBSET VS. CARDINAL MINIMALITY

In this subsection, we provide a more detailed discussion of the distinction between cardinal and subset minimality. Cardinal minimality offers a substantially stronger guarantee, as it corresponds to a globally minimal explanation size, whereas subset minimality only ensures a local form of minimality.

To see why cardinal minimality is stronger, note that if $S \subseteq [n]$ is a cardinally minimal explanation, then no smaller set $S'$ with $|S'| < |S|$ can qualify as an explanation. Hence, no strict subset $S' \subsetneq S$ is an explanation, which means $S$ is also subset minimal.

However, the reverse does not hold. Consider the function:

$$f := \mathbf{x}_1 \vee (\mathbf{x}_2 \wedge \mathbf{x}_3) \tag{13}$$

and the assignment $\mathbf{x} := (1, 1, 1)$, which gives $f(1, 1, 1) = 1$. Fixing feature $\mathbf{x}_1 = 1$ yields a cardinally minimal (and subset minimal) sufficient reason, since the prediction remains 1 regardless of $\mathbf{x}_2, \mathbf{x}_3$. But fixing both $\mathbf{x}_2 = 1$ and $\mathbf{x}_3 = 1$ also gives a subset minimal explanation—yet not a cardinally minimal one. Thus, while every cardinally minimal explanation is subset minimal, not every subset minimal explanation is cardinally minimal.

## C    PROOF OF PROPOSITION 1

**Proposition 1.** *While the non-probabilistic case ($\delta = 1$) admits a unique subset-minimal global sufficient reason (Bassan et al., 2024), in the general probabilistic setting (for arbitrary $\delta$), there exist functions $f$ that have $\Theta(\frac{2^n}{\sqrt{n}})$ subset-minimal global sufficient reasons.*

*Proof.* It is known that certain Boolean functions admit an exponential number of subset-minimal local (non-probabilistic) sufficient reasons for some input $\mathbf{x}$ Bassan et al. (2024). In particular, this phenomenon was shown to occur in functions of the following form. We demonstrate that the same function admits an exponential number of *global* and *probabilistic* sufficient reasons. Notably, the mentioned function is a special case of a broader class of *threshold Boolean functions* described in Wegener (2005). Specifically, for $n = 2k + 1$ with $k \in \mathbb{N}$, this function is defined as:

$$f(\mathbf{x}) := \begin{cases} 1 & \text{if } \sum_{i=1}^{n} \mathbf{x}_i \geq k + 1 \\ 0 & \text{otherwise} \end{cases} \tag{14}$$

where $\mathbf{x}$ is drawn from a uniform distribution $\mathcal{D}$. Notably, the function $f$ is *symmetric*, in the sense of symmetric threshold Boolean functions Wegener (2005): its output depends solely on the number of 1's (or, equivalently, 0's) in the input and is invariant under any permutation of input bits. Furthermore, the two extreme inputs — the all-zeros vector $(0, \ldots, 0)$ and the all-ones vector $(1, \ldots, 1)$ — each admit an exponential number of subset-minimal local (non-probabilistic) sufficient reasons.

Since the function $f$ is symmetric, the condition $v_{\text{suff}}^g(S_1) = v_{\text{suff}}^g(S_2)$ holds for any two subsets $S_1, S_2 \subseteq [n]$. Moreover, it can be verified that for any subset $S \subseteq [n]$ of size $k + 1$, we have:

$$(\forall i \in S). \quad v_{\text{suff}}^g(S) > v_{\text{suff}}^g(S \setminus \{i\}) \tag{15}$$

We also deliberately set $\delta$ to satisfy:

$$v_{\text{suff}}^g(S \setminus \{i\}) < \delta \leq v_{\text{suff}}^g(S), \tag{16}$$

and also set $k := \lfloor \frac{n}{2} \rfloor$. Each of these subsets is *subset-minimal* because any subset of size at most $\lfloor \frac{n}{2} \rfloor - 1$ fails to be a sufficient reason for $\langle f, \mathbf{x} \rangle$ (with respect to the $\delta$ threshold). Therefore, we can directly apply the same analysis as in (Bassan et al., 2024). In particular, there are exactly $\binom{n}{\lfloor \frac{n}{2} \rfloor}$ subset-minimal local sufficient reasons for $\langle f, \mathbf{x} \rangle$. Using Stirling's approximation, we obtain:

$$\lim_{n \to \infty} \frac{2\sqrt{2\pi}}{e^2} \cdot \frac{2^n}{\sqrt{n}} \leq \binom{n}{\lfloor \frac{n}{2} \rfloor} \leq \frac{e}{\pi} \cdot \frac{2^n}{\sqrt{n}} \tag{17}$$

This yields the corresponding bound on the number of subset-minimal global $\delta$-sufficient reasons.

$\square$

## D    PROOF OF PROPOSITION 2

**Proposition 2.** *While the local probabilistic setting (for any $\delta$) lacks monotonicity — i.e., the value functions $v_{con}^\ell$ and $v_{suff}^\ell$ are non-monotonic (Arenas et al., 2022; Izza et al., 2023; Subercaseaux et al., 2025; Izza et al., 2024b) — in the global probabilistic setting (also for any $\delta$), both value functions $v_{con}^g$ and $v_{suff}^g$ are monotonic non-decreasing.*

*Proof.* In this section, we prove the monotonicity property for both types of global explanations — sufficient and contrastive. We begin by establishing the property for global sufficiency. To build intuition, we first focus on the simpler case of Boolean functions, then generalize the result to functions with discrete multi-valued input domains and multiple output classes. We further extend the result to continuous input domains and show that the monotonicity property holds for any well-defined classification function. Finally, we demonstrate how the same monotonicity guarantees apply to global contrastive explanations. Importantly, none of our proofs rely on the assumption of feature independence — the monotonicity property holds strongly for *any* underlying distribution.

Lastly, we present a simple example showing that the monotonicity property does not hold for the local probabilistic sufficient or contrastive value functions.

**Lemma 1.** *The global sufficient value function $v_{suff}^g$ is monotonic non-decreasing for Boolean functions, under any data distribution.*

*Proof.* Given an arbitrary set $S \in [n]$, and a feature $i \notin S$, we have that:

$$\mathbf{Pr}(\mathbf{x}) = \mathbf{Pr}(\mathbf{x}_S, \mathbf{x}_{\bar{S}}) = \mathbf{Pr}(\mathbf{x}_{\bar{S}} \mid \mathbf{x}_S) \cdot \mathbf{Pr}(\mathbf{x}_S) \tag{18}$$

Consequently, this implies that:

$$\begin{aligned}
\mathbf{Pr}(\mathbf{x}) &= \mathbf{Pr}(\mathbf{x}_{\overline{S \cup \{i\}}} \mid \mathbf{x}_{S \cup \{i\}}) \cdot \mathbf{Pr}(\mathbf{x}_{S \cup \{i\}}) \\
&= \mathbf{Pr}(\mathbf{x}_{\overline{S \cup \{i\}}} \mid \mathbf{x}_{S \cup \{i\}}) \cdot \mathbf{Pr}(\mathbf{x}_S, x_i) \\
&= \mathbf{Pr}(\mathbf{x}_{\overline{S \cup \{i\}}} \mid \mathbf{x}_{S \cup \{i\}}) \cdot \mathbf{Pr}(x_i \mid \mathbf{x}_S) \cdot \mathbf{Pr}(\mathbf{x}_S)
\end{aligned} \tag{19}$$

Moreover, for any two points $\mathbf{x}, \mathbf{x}' \sim \mathcal{D}$ such that $\mathbf{x}_S = \mathbf{x}'_S$ and $f(\mathbf{x}) = 1 - f(\mathbf{x}')$, it holds that:

$$\mathbf{Pr}_{\mathbf{z} \sim \mathcal{D}}(f(\mathbf{z}) = f(\mathbf{x}) \mid \mathbf{z}_S = \mathbf{x}_S) = 1 - \mathbf{Pr}_{\mathbf{z} \sim \mathcal{D}}(f(\mathbf{z}) = f(\mathbf{x}') \mid \mathbf{z}_S = \mathbf{x}'_S) \tag{20}$$

To simplify notation, we occasionally omit the distribution notation $\sim \mathcal{D}$ in the local probability expression $\mathbf{Pr}_{\mathbf{z} \sim \mathcal{D}}(f(\mathbf{z}) = f(\mathbf{x}) \mid \mathbf{z}_S = \mathbf{x}_S)$, and use $f^+$ and $f^-$ to denote the events $f(\mathbf{z}) = 1$ and $f(\mathbf{z}) = 0$, respectively.

We begin with a technical simplification of the probability term to facilitate the proof, and then proceed to establish monotonicity.

**Simplifying the probability term.** Let $\mathcal{D}_S$ denote the distribution restricted to the set $S$. Given a point $\mathbf{x} \sim \mathcal{D}$ and a set $S$, consider all points $\mathbf{z} \sim \mathcal{D}$ such that $\mathbf{z}_S = \mathbf{x}_S$ and $f(\mathbf{z}) = b$, where $b \in \{0, 1\}$ — they all share the same local probability $\mathbf{Pr}_{\mathbf{z} \sim \mathcal{D}}(f(\mathbf{z}) = b \mid \mathbf{z}_S = \mathbf{x}_S)$. For every $\mathbf{x} \sim \mathcal{D}$ that shares the same $x_S$ and the same output $b$, we marginalize over $x_{\bar{S}}$. This yields $\mathbf{Pr}(f(\mathbf{x}) = b, \mathbf{x}_S) = \mathbf{Pr}(f(\mathbf{x}) = b \mid \mathbf{x}_S) \cdot \mathbf{Pr}(\mathbf{x}_S)$, from which we can infer:

$$\mathbf{Pr}_{\mathbf{z} \sim \mathcal{D}}(f(\mathbf{z}) = b \mid \mathbf{z}_S = \mathbf{x}_S) = \mathbf{Pr}(f(\mathbf{x}) = b \mid \mathbf{x}_S). \tag{21}$$

Thus, for $\mathbb{E}_{\mathbf{x} \sim \mathcal{D}}[\mathbf{Pr}_{\mathbf{z} \sim \mathcal{D}}(f(\mathbf{z}) = f(\mathbf{x}) \mid \mathbf{z}_S = \mathbf{x}_S)]$, we get that:

$$\begin{aligned}
&\mathbb{E}_{\mathbf{x} \sim \mathcal{D}}[\mathbf{Pr}_{\mathbf{z} \sim \mathcal{D}}(f(\mathbf{z}) = f(\mathbf{x}) \mid \mathbf{z}_S = \mathbf{x}_S)] \\
&= \sum_{\mathbf{x}_S \sim \mathcal{D}_S, f^+} \mathbf{Pr}(f^+, \mathbf{x}_S) \cdot \mathbf{Pr}_{\mathbf{z} \sim \mathcal{D}}(f^+ \mid \mathbf{z}_S = \mathbf{x}_S) + \sum_{\mathbf{x}_S \sim \mathcal{D}_S, f^-} \mathbf{Pr}(f^-, \mathbf{x}_S) \cdot \mathbf{Pr}_{\mathbf{z} \sim \mathcal{D}}(f^- \mid \mathbf{z}_S = \mathbf{x}_S) \\
&= \sum_{\mathbf{x}_S \sim \mathcal{D}_S} \mathbf{Pr}(\mathbf{x}_S) \cdot \left[ \left(\mathbf{Pr}_{\mathbf{z} \sim \mathcal{D}}(f^+ \mid \mathbf{z}_S = \mathbf{x}_S)\right)^2 + \left(\mathbf{Pr}_{\mathbf{z} \sim \mathcal{D}}(f^- \mid \mathbf{z}_S = \mathbf{x}_S)\right)^2 \right] \\
&= \sum_{\mathbf{x}_S \sim \mathcal{D}_S} \mathbf{Pr}(\mathbf{x}_S) \cdot \left[ \left(\mathbf{Pr}_{\mathbf{z} \sim \mathcal{D}}(f^+ \mid \mathbf{z}_S = \mathbf{x}_S)\right)^2 + \left(1 - \mathbf{Pr}_{\mathbf{z} \sim \mathcal{D}}(f^+ \mid \mathbf{z}_S = \mathbf{x}_S)\right)^2 \right],
\end{aligned} \tag{22}$$

From which it follows that:

$$\mathbb{E}_{\mathbf{x}\sim\mathcal{D}}[\mathbf{Pr}_{\mathbf{z}\sim\mathcal{D}}(f(\mathbf{z}) = f(\mathbf{x}) \mid \mathbf{z}_{S\cup\{t\}} = \mathbf{x}_{S\cup\{t\}})]$$

$$= \sum_{\mathbf{x}_{S\cup\{t\}}\sim\mathcal{D}_{S\cup\{t\}}} \mathbf{Pr}(\mathbf{x}_{S\cup\{t\}}) \cdot \left[\left(\mathbf{Pr}_{\mathbf{z}}(f^+ \mid \mathbf{z}_{S\cup\{t\}} = \mathbf{x}_{S\cup\{t\}})\right)^2 + \left(1 - \mathbf{Pr}_{\mathbf{z}}(f^+ \mid \mathbf{z}_{S\cup\{t\}} = \mathbf{x}_{S\cup\{t\}})\right)^2\right]$$

$$= \sum_{\substack{\mathbf{x}_S\sim\mathcal{D}_S \\ x_t=1}} \mathbf{Pr}(x_t = 1, \mathbf{x}_S) \cdot \left[\left(\mathbf{Pr}_{\mathbf{z}}(f^+ \mid \mathbf{z}_S = \mathbf{x}_S, z_t = 1)\right)^2 + \left(1 - \mathbf{Pr}_{\mathbf{z}}(f^+ \mid \mathbf{z}_S = \mathbf{x}_S, z_t = 1)\right)^2\right] +$$

$$\sum_{\substack{\mathbf{x}_S\sim\mathcal{D}_S \\ x_t=0}} \mathbf{Pr}(x_t = 0, \mathbf{x}_S) \cdot \left[\left(\mathbf{Pr}_{\mathbf{z}}(f^+ \mid \mathbf{z}_S = \mathbf{x}_S, z_t = 0)\right)^2 + \left(1 - \mathbf{Pr}_{\mathbf{z}}(f^+ \mid \mathbf{z}_S = \mathbf{x}_S, z_t = 0)\right)^2\right]$$

$$(23)$$

**Proving monotonicity.** We now proceed to prove the monotonicity claim, building on the previous simplification. We begin by introducing a few additional notations. Let $g_1^+ := \mathbf{Pr}_{\mathbf{z}\sim\mathcal{D}}(f(\mathbf{z}) = 1 \mid \mathbf{z}_S = \mathbf{x}_S, z_t = 1)$ and $g_0^+ := \mathbf{Pr}_{\mathbf{z}\sim\mathcal{D}}(f(\mathbf{z}) = 1 \mid \mathbf{z}_S = \mathbf{x}_S, z_t = 0)$. Similarly, define $P_1 := \mathbf{Pr}(z_t = 1 \mid \mathbf{x}_S)$ and $P_0 := \mathbf{Pr}(z_t = 0 \mid \mathbf{x}_S)$. Then, the following expectations $\mathbb{E}_{\mathbf{x}\sim\mathcal{D}}[\mathbf{Pr}_{\mathbf{z}\sim\mathcal{D}}(f(\mathbf{z}) = f(\mathbf{x}) \mid \mathbf{z}_S = \mathbf{x}_S)]$ and $\mathbb{E}_{\mathbf{x}\sim\mathcal{D}}[\mathbf{Pr}_{\mathbf{z}\sim\mathcal{D}}(f(\mathbf{z}) = f(\mathbf{x}) \mid \mathbf{z}_{S\cup\{t\}} = \mathbf{x}_{S\cup\{t\}})]$ can be simplified accordingly [1]:

$$\mathbb{E}_{\mathbf{x}\sim\mathcal{D}}[\mathbf{Pr}_{\mathbf{z}\sim\mathcal{D}}(f(\mathbf{z}) = f(\mathbf{x}) \mid \mathbf{z}_S = \mathbf{x}_S)]$$

$$= \sum_{\mathbf{x}_S\sim\mathcal{D}_S} \mathbf{Pr}(\mathbf{x}_S) \cdot [(P_1 \cdot g_1^+ + P_0 \cdot g_0^+)^2 + (1 - (P_1 \cdot g_1^+ + P_0 \cdot g_0^+))^2]$$

$$= \sum_{\mathbf{x}_S\sim\mathcal{D}_S} \mathbf{Pr}(\mathbf{x}_S) \cdot [2(P_1 \cdot g_1^+ + P_0 \cdot g_0^+)^2 - 2(P_1 \cdot g_1^+ + P_0 \cdot g_0^+) + 1].$$

$$(24)$$

As a result, we obtain the following:

$$\mathbb{E}_{\mathbf{x}\sim\mathcal{D}}[\mathbf{Pr}_{\mathbf{z}\sim\mathcal{D}}(f(\mathbf{z}) = f(\mathbf{x}) \mid \mathbf{z}_{S\cup\{t\}} = \mathbf{x}_{S\cup\{t\}})]$$

$$= \sum_{\substack{\mathbf{x}_S\sim\mathcal{D}_S \\ x_t=1}} \mathbf{Pr}(\mathbf{x}_S) \cdot P_1 \cdot ((g_1^+)^2 + (1 - g_1^+)^2) + \sum_{\substack{\mathbf{x}_S\sim\mathcal{D}_S \\ x_t=0}} \mathbf{Pr}(\mathbf{x}_S) \cdot P_0 \cdot ((g_0^+)^2 + (1 - g_0^+)^2)$$

$$= \sum_{\substack{\mathbf{x}_S\sim\mathcal{D}_S \\ x_t=1}} \mathbf{Pr}(\mathbf{x}_S) \cdot P_1 \cdot (2(g_1^+)^2 - 2g_1^+ + 1) + \sum_{\substack{\mathbf{x}_S\sim\mathcal{D}_S \\ x_t=0}} \mathbf{Pr}(\mathbf{x}_S) \cdot P_0 \cdot (2(g_0^+)^2 - 2g_0^+ + 1)$$

$$= \sum_{\mathbf{x}_S\sim\mathcal{D}_S} \mathbf{Pr}(\mathbf{x}_S) \cdot [P_1 \cdot (2(g_1^+)^2 - 2g_1^+ + 1) + P_0 \cdot (2(g_0^+)^2 - 2g_0^+ + 1)]$$

$$= \sum_{\mathbf{x}_S\sim\mathcal{D}_S} \mathbf{Pr}(\mathbf{x}_S) \cdot [2P_1 \cdot ((g_1^+)^2 - g_1^+) + 2P_0 \cdot ((g_0^+)^2 - g_0^+) + 1]$$

$$= \sum_{\mathbf{x}_S\sim\mathcal{D}_S} \mathbf{Pr}(\mathbf{x}_S) \cdot [2(P_1 \cdot (g_1^+)^2 + P_0 \cdot (g_0^+)^2) - 2(P_1 \cdot g_1^+ + P_0 \cdot g_0^+) + 1].$$

$$(25)$$

Let $\Delta_t(S, f)$ denote $\mathbb{E}_{\mathbf{x}\sim\mathcal{D}}[\mathbf{Pr}_{\mathbf{z}\sim\mathcal{D}}(f(\mathbf{z}) = f(\mathbf{x}) \mid \mathbf{z}_{S\cup\{t\}} = \mathbf{x}_{S\cup\{t\}})] - \mathbb{E}_{\mathbf{x}\sim\mathcal{D}}[\mathbf{Pr}_{\mathbf{z}\sim\mathcal{D}}(f(\mathbf{z}) = f(\mathbf{x}) \mid \mathbf{z}_S = \mathbf{x}_S)]$. Without loss of generality, we assume that both $P_1$ and $P_0$ are positive. Otherwise, it can be verified that $\Delta_t(S, f) = 0$. Then, we get:

---

[1] Note that the notations $P_1$, $P_0$, $g_1^+$, and $g_0^+$—as well as all others used throughout the paper—are not fixed constants, but rather depend on the specific input $\mathbf{x}_S$. Formally, they should be written as $P_1(\mathbf{x}_S)$, $P_0(\mathbf{x}_S)$, and so on. However, for readability, we omit the explicit dependence on $\mathbf{x}_S$.

$$\Delta_t(S, f)$$
$$= \sum_{\mathbf{x}_S \sim \mathcal{D}_S} \mathbf{Pr}(\mathbf{x}_S) \cdot [2(P_1 \cdot (g_1^+)^2 + P_0 \cdot (g_0^+)^2) - 2(P_1 \cdot g_1^+ + P_0 \cdot g_0^+)^2]$$
$$= \sum_{\mathbf{x}_S \sim \mathcal{D}_S} 2 \cdot \mathbf{Pr}(\mathbf{x}_S) \cdot [(P_1 - P_1^2)(g_1^+)^2 + (P_0 - P_0^2)(g_0^+)^2 - 2 \cdot P_1 \cdot P_0 \cdot g_1^+ \cdot g_0^+] \quad (26)$$
$$= \sum_{\mathbf{x}_S \sim \mathcal{D}_S} 2 \cdot \mathbf{Pr}(\mathbf{x}_S) \cdot P_1 \cdot P_0 \cdot (g_1^+ - g_0^+)^2$$
$$\geq 0.$$

Note that $P_1 - P_1^2 = P_1(1 - P_1) = P_1 P_0$ and similarly: $P_0 - P_0^2 = P_0(1 - P_0) = P_0 P_1)$. Using these identities along with the previous result, we conclude that $\mathbb{E}_{\mathbf{x} \sim \mathcal{D}}[\mathbf{Pr}_{\mathbf{z} \sim \mathcal{D}}(f(\mathbf{z}) = f(\mathbf{x}) \mid \mathbf{z}_S = \mathbf{x}_S)]$ is monotone, thereby concluding our proof.

$\square$

**Lemma 2.** *The global sufficient value function $v_{suff}^g$ is monotonic non-decreasing for functions with discrete multi-valued input domains and multiple output classes, under any data distribution.*

*Proof.* Extending the proof from the simplified Boolean case, we now generalize the result to functions whose input domains and output ranges each consist of a finite set of values. Let $\mathcal{K}$ represent the set of output classes.

We can generalize equation 22 as follows:

$$\mathbb{E}_{\mathbf{x} \sim \mathcal{D}}[\mathbf{Pr}_{\mathbf{z} \sim \mathcal{D}}(f(\mathbf{z}) = f(\mathbf{x}) \mid \mathbf{z}_S = \mathbf{x}_S)] =$$
$$\sum_{\mathbf{x}_S \sim \mathcal{D}_S} \mathbf{Pr}(\mathbf{x}_S) \cdot \Big[ \sum_{j \in \mathcal{K}} \big( \mathbf{Pr}_{\mathbf{z} \sim \mathcal{D}}(f(\mathbf{z}) = j \mid \mathbf{z}_S = \mathbf{x}_S) \big)^2 \Big] \quad (27)$$

Given the condition $\mathbf{z}_S = \mathbf{x}_S$, suppose feature $t$ can take $r$ possible values $\{v_1, \ldots, v_r\}$. Then, equation 23 can be generalized as follows:

$$\mathbb{E}_{\mathbf{x} \sim \mathcal{D}}[\mathbf{Pr}_{\mathbf{z} \sim \mathcal{D}}(f(\mathbf{z}) = f(\mathbf{x}) \mid \mathbf{z}_{S \cup \{t\}} = \mathbf{x}_{S \cup \{t\}})]$$
$$= \sum_{k=1}^{r} \Big[ \sum_{\substack{\mathbf{x}_S \sim \mathcal{D}_S \\ x_t = v_k}} \mathbf{Pr}(\mathbf{x}_S) \cdot \mathbf{Pr}(z_t = v_k \mid \mathbf{x}_S) \cdot \Big[ \sum_{j \in \mathcal{K}} \big( \mathbf{Pr}_{\mathbf{z} \sim \mathcal{D}}(f(\mathbf{z}) = j \mid \mathbf{z}_S = \mathbf{x}_S, z_t = v_k) \big)^2 \Big] \Big].$$

$$(28)$$

Let $P_k$ denote $\mathbf{Pr}(z_t = v_k \mid \mathbf{x}_S)$, Let $g_k^j$ denote $\mathbf{Pr}_{\mathbf{z} \sim \mathcal{D}}(f(\mathbf{z}) = j \mid \mathbf{z}_S = \mathbf{x}_S, z_t = v_k)$, that is, $z_t$ takes the value $v_k$ and the output class is $j$. Then, we have

$$\mathbb{E}_{\mathbf{x} \sim \mathcal{D}}[\mathbf{Pr}_{\mathbf{z} \sim \mathcal{D}}(f(\mathbf{z}) = f(\mathbf{x}) \mid \mathbf{z}_S = \mathbf{x}_S)] = \sum_{\mathbf{x}_S \sim \mathcal{D}_S} \mathbf{Pr}(\mathbf{x}_S) \cdot \Big[ \sum_{j \in \mathcal{K}} \big( \sum_{k=1}^{r} P_k \cdot g_k^j \big)^2 \Big] \quad (29)$$

and

$$\mathbb{E}_{\mathbf{x} \sim \mathcal{D}}[\mathbf{Pr}_{\mathbf{z} \sim \mathcal{D}}(f(\mathbf{z}) = f(\mathbf{x}) \mid \mathbf{z}_{S \cup \{t\}} = \mathbf{x}_{S \cup \{t\}})] = \sum_{\mathbf{x}_S \sim \mathcal{D}_S} \mathbf{Pr}(\mathbf{x}_S) \cdot \Big[ \sum_{k=1}^{r} P_k \cdot \big( \sum_{j \in \mathcal{K}} (g_k^j)^2 \big) \Big]$$

Combining these two implications, we can conclude that:

$$\Delta_t(S, f)$$

$$= \sum_{\mathbf{x}_S \sim \mathcal{D}_S} \mathbf{Pr}(\mathbf{x}_S) \cdot \Big[ \sum_{k=1}^{r} P_k \cdot \big( \sum_{j \in \mathcal{K}} (g_k^j)^2 \big) - \sum_{j \in \mathcal{K}} \big( \sum_{k=1}^{r} P_k \cdot g_k^j \big)^2 \Big]$$

$$= \sum_{\mathbf{x}_S \sim \mathcal{D}_S} \mathbf{Pr}(\mathbf{x}_S) \cdot \Big[ \sum_{j \in \mathcal{K}} \big( \sum_{k=1}^{r} P_k \cdot (g_k^j)^2 \big) - \sum_{j \in \mathcal{K}} \big( \sum_{k=1}^{r} (P_k \cdot g_k^j)^2 + \sum_{k<l} 2 \cdot P_k \cdot P_l \cdot g_k^j \cdot g_l^j \big) \Big] \quad (30)$$

$$= \sum_{\mathbf{x}_S \sim \mathcal{D}_S} \mathbf{Pr}(\mathbf{x}_S) \cdot \Big[ \sum_{j \in \mathcal{K}} \sum_{k \neq l} P_k \cdot P_l \cdot (g_k^j)^2 - \sum_{j \in \mathcal{K}} \sum_{k<l} 2 \cdot P_k \cdot P_l \cdot g_k^j \cdot g_l^j \Big]$$

$$= \sum_{\mathbf{x}_S \sim \mathcal{D}_S} \mathbf{Pr}(\mathbf{x}_S) \cdot \sum_{j \in \mathcal{K}} \Big[ \sum_{k<l} P_k \cdot P_l \cdot (g_k^j - g_l^j)^2 \Big]$$

$$\geq 0.$$

Hence, $\mathbb{E}_{\mathbf{x} \sim \mathcal{D}}[\mathbf{Pr}_{\mathbf{z} \sim \mathcal{D}}(f(\mathbf{z}) = f(\mathbf{x}) \mid \mathbf{z}_S = \mathbf{x}_S)]$ is monotone, which completes the proof for this setting.

$\square$

**Lemma 3.** *The global sufficient value function $v_{suff}^g$ is monotonic non-decreasing for functions with continuous input domains and multiple output classes, under any data distribution.*

*Proof.* We extend the monotonicity results to functions where features are defined over continuous domains. One approach is to partition the domain of the continuous features into two disjoint subdomains and then adapt the proofs in lemma 2 by revising the notation accordingly. The core reasoning remains unchanged when transitioning from the discrete to the continuous case—only the notation differs.

Alternatively, we provide a direct proof below. Assume all features lie in continuous domains, and let $p(\mathbf{x}_S, \mathbf{x}_{\bar{S}})$ denote the corresponding joint probability density function. Moreover, let $p(x_i)$ denote the marginal probability density function of $x_i$, and $\mathcal{X}_i$ denote the domain of feature $i$.

We define marginal density as

$$p(\mathbf{x}_S) := \int_{\mathbf{x}_{\bar{S}} \in \mathcal{X}_{\bar{S}}} p(\mathbf{x}_S, \mathbf{x}_{\bar{S}}) d\mathbf{x}_{\bar{S}}, \quad (31)$$

where $\mathcal{X}_{\bar{S}}$ represents the domain of features in $\bar{S}$.

And the conditional density is defined as:

$$p(x_t \mid \mathbf{x}_S) := \frac{p(\mathbf{x}_S, x_t)}{p(\mathbf{x}_S)}. \quad (32)$$

The local probability of $S$ given $\mathbf{x}$ is defined as

$$\mathbf{Pr}_{\mathbf{z} \sim \mathcal{D}}(f(\mathbf{z}) = j \mid \mathbf{z}_S = \mathbf{x}_S) = \frac{\int_{\mathbf{z}_{\bar{S}} \in \mathcal{X}_{\bar{S}}} I(f(\mathbf{x}_S, \mathbf{z}_{\bar{S}}) = j) \cdot p(\mathbf{x}_S, \mathbf{z}_{\bar{S}}) d\mathbf{z}_{\bar{S}}}{p(\mathbf{x}_S)}, \quad (33)$$

This term can also be decomposed into:

$$\mathbf{Pr}_{\mathbf{z} \sim \mathcal{D}}(f(\mathbf{z}) = j \mid \mathbf{z}_S = \mathbf{x}_S) = \int_{z_t \in \mathcal{X}_t} p(z_t \mid \mathbf{x}_S) \cdot \mathbf{Pr}_{\mathbf{z} \sim \mathcal{D}}(f(\mathbf{z}) = j \mid \mathbf{z}_S = \mathbf{x}_S, z_t = x_t) dz_t \quad (34)$$

via an additional feature $t \notin S$.

We then modify equation 27 to

$$\mathbb{E}_{\mathbf{x} \sim \mathcal{D}}[\mathbf{Pr}_{\mathbf{z} \sim \mathcal{D}}(f(\mathbf{z}) = f(\mathbf{x}) \mid \mathbf{z}_S = \mathbf{x}_S)]$$

$$= \int_{\mathbf{x}_S \in \mathcal{X}_S} p(\mathbf{x}_S) \cdot \Big[ \sum_{j \in \mathcal{K}} \big( \mathbf{Pr}_{\mathbf{z} \sim \mathcal{D}}(f(\mathbf{z}) = j \mid \mathbf{z}_S = \mathbf{x}_S) \big)^2 \Big] d\mathbf{x}_S, \quad (35)$$

while equation 28 is changed to

$$\mathbb{E}_{\mathbf{x}\sim\mathcal{D}}[\mathbf{Pr}_{\mathbf{z}\sim\mathcal{D}}(f(\mathbf{z}) = f(\mathbf{x}) \mid \mathbf{z}_{S\cup\{t\}} = \mathbf{x}_{S\cup\{t\}})]$$
$$= \int_{\mathbf{x}_S\in\mathcal{X}_S} p(\mathbf{x}_S)\cdot\Big[\int_{z_t\in\mathcal{X}_t} p(z_t\mid\mathbf{x}_S)\cdot\Big(\sum_{j\in\mathcal{K}}\big(\mathbf{Pr}_{\mathbf{z}\sim\mathcal{D}}(f(\mathbf{z}) = j\mid\mathbf{z}_{S\cup\{t\}} = \mathbf{x}_{S\cup\{t\}})\big)^2\Big)dz_t\Big]d\mathbf{x}_S. \tag{36}$$

For simplicity, let $Y_j(t)$ denote $\mathbf{Pr}_{\mathbf{z}\sim\mathcal{D}}(f(\mathbf{z}) = j \mid \mathbf{z}_S = \mathbf{x}_S, z_t = v_t)$, and $p_{t|S}$ denote $p(z_t \mid \mathbf{x}_S)$. Note that both $Y_j(t)$ and $p_{t|S}$ are associated with a fixed $\mathbf{x}_S$ [2]. We prove monotonicity as follows:

$$\Delta_t(S, f)$$
$$= \mathbb{E}_{\mathbf{x}\sim\mathcal{D}}[\mathbf{Pr}_{\mathbf{z}\sim\mathcal{D}}(f(\mathbf{z}) = f(\mathbf{x}) \mid \mathbf{z}_{S\cup\{t\}} = \mathbf{x}_{S\cup\{t\}})] - \mathbb{E}_{\mathbf{x}\sim\mathcal{D}}[\mathbf{Pr}_{\mathbf{z}\sim\mathcal{D}}(f(\mathbf{z}) = f(\mathbf{x}) \mid \mathbf{z}_S = \mathbf{x}_S)]$$
$$= \int_{\mathbf{x}_S\in\mathcal{X}_S} p(\mathbf{x}_S)\cdot\Big[\int_{z_t\in\mathcal{X}_t} p_{t|S}\cdot\Big(\sum_{j\in\mathcal{K}}\big(\mathbf{Pr}_{\mathbf{z}\sim\mathcal{D}}(f(\mathbf{z}) = j\mid\mathbf{z}_{S\cup\{t\}} = \mathbf{x}_{S\cup\{t\}})\big)^2\Big)dz_t\Big]d\mathbf{x}_S -$$
$$\int_{\mathbf{x}_S\in\mathcal{X}_S} p(\mathbf{x}_S)\cdot\Big[\sum_{j\in\mathcal{K}}\Big(\int_{z_t\in\mathcal{X}_t} p_{t|S}\cdot\mathbf{Pr}_{\mathbf{z}\sim\mathcal{D}}(f(\mathbf{z}) = j\mid\mathbf{z}_S = \mathbf{x}_S, z_t = x_t)dz_t\Big)^2\Big]d\mathbf{x}_S$$
$$= \int_{\mathbf{x}_S\in\mathcal{X}_S} p(\mathbf{x}_S)\cdot\Big[\sum_{j\in\mathcal{K}}\Big(\int_{\mathcal{X}_t} p_{t|S}\cdot Y_j(t)^2 dz_t - \big(\int_{\mathcal{X}_t} p_{t|S}\cdot Y_j(t)dz_t\big)^2\Big)\Big]d\mathbf{x}_S$$
$$= \int_{\mathbf{x}_S\in\mathcal{X}_S} p(\mathbf{x}_S)\cdot\Big[\sum_{j\in\mathcal{K}}\frac{2}{2}\cdot\Big(\int_{\mathcal{X}_t}\int_{\mathcal{X}_{t'}} p_{t|S}\cdot p_{t'|S}\cdot\big(Y_j(t)^2 - Y_j(t)Y_j(t')\big)dz_tdz_{t'}\Big)\Big]d\mathbf{x}_S$$
$$= \int_{\mathbf{x}_S\in\mathcal{X}_S} p(\mathbf{x}_S)\cdot\Big[\sum_{j\in\mathcal{K}}\frac{1}{2}\cdot\Big(\int_{\mathcal{X}_t}\int_{\mathcal{X}_{t'}} p_{t|S}\cdot p_{t'|S}\cdot\big(Y_j(t) - Y_j(t')\big)^2 dz_tdz_{t'}\Big)\Big]d\mathbf{x}_S$$
$$\geq 0. \tag{37}$$

$\square$

**Proposition 2.** *While the local probabilistic setting (for any $\delta$) lacks monotonicity — i.e., the value functions $v_{con}^\ell$ and $v_{suff}^\ell$ are non-monotonic (Arenas et al., 2022; Izza et al., 2023; Subercaseaux et al., 2025; Izza et al., 2024b) — in the global probabilistic setting (also for any $\delta$), both value functions $v_{con}^g$ and $v_{suff}^g$ are monotonic non-decreasing.*

*Proof.* As a consequence of lemmas 1 to 3, we conclude that the global sufficient value function $v_{suff}^g$ is monotonic non-decreasing for any well-defined classification function — whether its features are Boolean, discrete, continuous, or a combination — under any data distribution.

$$v_{con}^g(S) = 1 - v_{suff}^g(\bar{S}) \tag{38}$$

This implies that $v_{con}^g$ is monotonic non-increasing with respect to adding features to $\bar{S}$. Equivalently, $v_{con}^g$ is monotonic non-decreasing as features are removed from $\bar{S}$ and added to $S$, thus completing the proof.

$\square$

## E  PROOF OF PROPOSITION 3

**Proposition 3.** *While the local probabilistic sufficient setting (for any $\delta$) lacks supermodularity — even when $\mathcal{D}$ is uniform, i.e., the value function $v_{suff}^\ell$ is not supermodular — in the global probabilistic setting (also for any $\delta$), when $\mathcal{D}$ exhibits feature independence, the value function $v_{suff}^g$ is supermodular.*

*Proof.* This section establishes the supermodularity property for both global sufficient and contrastive explanations. As with monotonicity, we begin with Boolean functions and progressively generalize

---

[2]Changing $\mathbf{x}_S$ alters their values, we omit $\mathbf{x}_S$ in the notation for brevity.

to multi-valued discrete and continuous input domains, and finally consider the case of well-defined classification functions. Unlike the monotonicity property, however, these proofs require the assumption of feature independence. We also provide a counterexample to show that supermodularity may fail when this assumption is relaxed.

**Lemma 4.** *The global sufficient value function $v_{suff}^g$ is supermodular for Boolean functions, under the assumption of feature independence.*

*Proof.* Particularly, we will prove that:

$$\Delta_t(S, f) \leq \Delta_t(S', f), \tag{39}$$

where $S \subseteq S'$ and $t \notin S'$. That is, $\Delta_t(S, f)$ is monotone.

Clearly, if $S' = S$, then $\Delta_t(S, f) \leq \Delta_t(S', f)$ holds. Let $S' = S \cup \{i\}$, where $i \notin S$ and $i \neq t$. For the local probability $\mathbf{Pr}_{\mathbf{z} \sim \mathcal{D}}(f(\mathbf{z}) = f(\mathbf{x}) \mid \mathbf{z}_S = \mathbf{x}_S)$, under the assumption of feature independence, we have that the following holds:

$$
\begin{aligned}
\mathbf{Pr}_{\mathbf{z}}&(f(\mathbf{z}) = f(\mathbf{x}) \mid \mathbf{z}_S = \mathbf{x}_S) \\
&= \mathbf{Pr}(z_i = 1) \cdot \mathbf{Pr}(z_t = 1) \cdot \mathbf{Pr}_{\mathbf{z}}(f(\mathbf{z}) = f(\mathbf{x}) \mid \mathbf{z}_S = \mathbf{x}_S, z_i = 1, z_t = 1) + \\
&\quad \mathbf{Pr}(z_i = 1) \cdot \mathbf{Pr}(z_t = 0) \cdot \mathbf{Pr}_{\mathbf{z}}(f(\mathbf{z}) = f(\mathbf{x}) \mid \mathbf{z}_S = \mathbf{x}_S, z_i = 1, z_t = 0) + \\
&\quad \mathbf{Pr}(z_i = 0) \cdot \mathbf{Pr}(z_t = 1) \cdot \mathbf{Pr}_{\mathbf{z}}(f(\mathbf{z}) = f(\mathbf{x}) \mid \mathbf{z}_S = \mathbf{x}_S, z_i = 0, z_t = 1) + \\
&\quad \mathbf{Pr}(z_i = 0) \cdot \mathbf{Pr}(z_t = 0) \cdot \mathbf{Pr}_{\mathbf{z}}(f(\mathbf{z}) = f(\mathbf{x}) \mid \mathbf{z}_S = \mathbf{x}_S, z_i = 0, z_t = 0).
\end{aligned} \tag{40}
$$

Let $P_1$, $P_0$, $Q_1$, and $Q_0$ denote $\mathbf{Pr}(z_t = 1)$, $\mathbf{Pr}(z_t = 0)$, $\mathbf{Pr}(z_i = 1)$, and $\mathbf{Pr}(z_i = 0)$, respectively. Moreover, let $g_{11}^+$, $g_{10}^+$, $g_{01}^+$, and $g_{00}^+$ denote $\mathbf{Pr}_{\mathbf{z} \sim \mathcal{D}}(f(\mathbf{z}) = 1 \mid \mathbf{z}_S = \mathbf{x}_S, z_i = 1, z_t = 1)$, $\mathbf{Pr}_{\mathbf{z} \sim \mathcal{D}}(f(\mathbf{z}) = 1 \mid \mathbf{z}_S = \mathbf{x}_S, z_i = 1, z_t = 0)$, $\mathbf{Pr}_{\mathbf{z} \sim \mathcal{D}}(f(\mathbf{z}) = 1 \mid \mathbf{z}_S = \mathbf{x}_S, z_i = 0, z_t = 1)$, and $\mathbf{Pr}_{\mathbf{z} \sim \mathcal{D}}(f(\mathbf{z}) = 1 \mid \mathbf{z}_S = \mathbf{x}_S, z_i = 0, z_t = 0)$, respectively.

We further decompose equation 26 into:

$$
\begin{aligned}
\Delta_t(S, f) &= \sum_{\mathbf{x}_S \sim \mathcal{D}_S} 2 \cdot \mathbf{Pr}(\mathbf{x}_S) \cdot P_1 P_0 \cdot \left[ (Q_1 \cdot g_{11}^+ + Q_0 \cdot g_{01}^+) - (Q_1 \cdot g_{10}^+ + Q_0 \cdot g_{00}^+) \right]^2 \\
&= \sum_{\mathbf{x}_S \sim \mathcal{D}_S} 2 \cdot \mathbf{Pr}(\mathbf{x}_S) \cdot P_1 P_0 \cdot \left[ Q_1 \cdot (g_{11}^+ - g_{10}^+) + Q_0 \cdot (g_{01}^+ - g_{00}^+) \right]^2.
\end{aligned} \tag{41}
$$

Similarly, the following also holds:

$$
\Delta_t(S \cup \{i\}, f) = \sum_{\mathbf{x}_S \sim \mathcal{D}_S} 2 \cdot \mathbf{Pr}(\mathbf{x}_S) \cdot P_1 P_0 \cdot \left[ Q_1 \cdot (g_{11}^+ - g_{10}^+)^2 + Q_0 \cdot (g_{01}^+ - g_{00}^+)^2 \right]. \tag{42}
$$

Thus, we get

$$
\begin{aligned}
\Delta_t(S \cup \{i\}, f) - \Delta_t(S, f) &= \sum_{\mathbf{x}_S \sim \mathcal{D}_S} 2 \cdot \mathbf{Pr}(\mathbf{x}_S) \cdot P_1 P_0 \cdot Q_1 Q_0 \cdot (g_{11}^+ - g_{10}^+ - g_{01}^+ + g_{00}^+)^2 \\
&\geq 0.
\end{aligned} \tag{43}
$$

This implies that $\Delta_t(S, f)$ is monotone, that is, $c(S \cup \{t\}, f) - c(S, f) \leq c(S' \cup \{t\}, f) - c(S', f)$. □

**Lemma 5.** *The global sufficient value function $v_{suff}^g$ is supermodular for functions with discrete multi-valued input domains and multiple output classes, under the assumption of feature independence.*

*Proof.* We extend the results of the previous section to multi-valued discrete functions, again with the assumption of feature independence. We use $P_k$ to denote $\mathbf{Pr}(z_t = v_k)$, and $Q_{k'}$ to denote $\mathbf{Pr}(z_i = v_k')$. Moreover, we use $g_{k'k}^j$ to denote $\mathbf{Pr}_{\mathbf{z} \sim \mathcal{D}}(f(\mathbf{z}) = j \mid \mathbf{z}_S = \mathbf{x}_S, z_i = v_k', z_t = v_k)$. We

decompose equation 30 to

$$\Delta_t(S, f) = \sum_{\mathbf{x}_S \sim \mathcal{D}_S} \mathbf{Pr}(\mathbf{x}_S) \cdot \sum_{j \in \mathcal{K}} \sum_{k<l} \Big[ \sum_{k'=1}^{r'} P_k P_l \cdot \Big( \sum_{k'=1}^{r'} Q_{k'} \cdot g_{k'k}^j - \sum_{k'=1}^{r'} Q_{k'} \cdot g_{k'l}^j \Big)^2 \Big]$$
$$= \sum_{\mathbf{x}_S \sim \mathcal{D}_S} \mathbf{Pr}(\mathbf{x}_S) \cdot \sum_{j \in \mathcal{K}} \sum_{k<l} \Big[ \sum_{k'=1}^{r'} P_k P_l \cdot \Big( \sum_{k'=1}^{r'} Q_{k'} \cdot (g_{k'k}^j - g_{k'l}^j) \Big)^2 \Big]. \tag{44}$$

Likewise, we have

$$\Delta_t(S \cup \{i\}, f) = \sum_{k'=1}^{r'} \Big[ \sum_{\substack{\mathbf{x}_S \sim \mathcal{D}_S \\ x_i = v_k'}} \mathbf{Pr}(\mathbf{x}_S) \cdot \mathbf{Pr}(z_i = v_k') \cdot \sum_{j \in \mathcal{K}} \sum_{k<l} \Big( \sum P_k P_l \cdot (g_{k'k}^j - g_{k'l}^j)^2 \Big) \Big]$$
$$= \sum_{\mathbf{x}_S \sim \mathcal{D}_S} \mathbf{Pr}(\mathbf{x}_S) \cdot \Big[ \sum_{k'=1}^{r'} Q_{k'} \cdot \sum_{j \in \mathcal{K}} \sum_{k<l} \Big( \sum P_k P_l \cdot (g_{k'k}^j - g_{k'l}^j)^2 \Big) \Big] \tag{45}$$
$$= \sum_{\mathbf{x}_S \sim \mathcal{D}_S} \mathbf{Pr}(\mathbf{x}_S) \cdot \sum_{j \in \mathcal{K}} \sum_{k<l} \Big[ \sum P_k P_l \cdot \sum_{k'=1}^{r'} Q_{k'} \cdot (g_{k'k}^j - g_{k'l}^j)^2 \Big].$$

Therefore, we get

$$\Delta_t(S \cup \{i\}, f) - \Delta_t(S, f)$$
$$= \sum_{\mathbf{x}_S \sim \mathcal{D}_S} \mathbf{Pr}(\mathbf{x}_S) \cdot \sum_{j \in \mathcal{K}} \Big[ \sum_{k<l} \sum_{k'<l'} P_k P_l Q_{k'} Q_{l'} \cdot (g_{k'k}^j - g_{k'l}^j - g_{l'k}^j + g_{l'l}^j)^2 \Big] \tag{46}$$
$$\geq 0.$$

$\square$

**Proposition 3.** *While the local probabilistic sufficient setting (for any δ) lacks supermodularity — even when $\mathcal{D}$ is uniform, i.e., the value function $v_{suff}^{\ell}$ is not supermodular — in the global probabilistic setting (also for any δ), when $\mathcal{D}$ exhibits feature independence, the value function $v_{suff}^g$ is supermodular.*

*Proof.* We can generalize the supermodularity results established in Lemma 5 to the continuous case, assuming that $\mathcal{D}_p$ satisfies feature independence. One approach is to partition the domain of the continuous features into two disjoint subdomains and then adapt the proofs in Lemma 5 by revising the notation accordingly.

Alternatively, we provide a direct proof below. Under this assumption, the marginal and conditional densities simplify as follows:

$$p(\mathbf{x}_S) = \prod_{i \in S} p(x_i), \text{ and } p(x_t \mid \mathbf{x}_S) = p(x_t). \tag{47}$$

Consequently, the local probability of $S$ conditioned on $\mathbf{x}$ becomes:

$$\mathbf{Pr}_{\mathbf{z} \sim \mathcal{D}}(f(\mathbf{z}) = j \mid \mathbf{z}_S = \mathbf{x}_S) = \int_{\mathbf{z}_{\bar{S}} \in \mathcal{X}_{\bar{S}}} I(f(\mathbf{x}_S, \mathbf{z}_{\bar{S}}) = j) \cdot \prod_{i \in \bar{S}} p(z_i) d\mathbf{z}_{\bar{S}}. \tag{48}$$

We let $Y_j(i, t)$ denote $\mathbf{Pr}_{\mathbf{z} \sim \mathcal{D}}(f(\mathbf{z}) = j \mid \mathbf{z}_S = \mathbf{x}_S, z_i = x_i, z_t = x_t)$, where $i \notin S, t \notin S$ and $i \neq t$. Evidently,

$$Y_j(i, t) = Y_j(t, i), \text{ and } Y_j(t) = \int_{\mathcal{X}_i} p_i \cdot Y_j(i, t) dz_i.$$

We have:

$$\Delta_t(S, f)$$

$$= \int_{\mathbf{x}_S \in \mathcal{X}_S} p(\mathbf{x}_S) \cdot \Big[ \sum_{j \in \mathcal{K}} \frac{1}{2} \Big( \int_{\mathcal{X}_t} \int_{\mathcal{X}_{t'}} p_t p_{t'} \big( Y_j(t) - Y_j(t') \big)^2 dz_t dz_{t'} \Big) \Big] d\mathbf{x}_S$$

$$= \int_{\mathbf{x}_S \in \mathcal{X}_S} p(\mathbf{x}_S) \cdot \Big[ \sum_{j \in \mathcal{K}} \frac{1}{2} \Big( \int_{\mathcal{X}_t} \int_{\mathcal{X}_{t'}} p_t p_{t'} \big( \int_{\mathcal{X}_i} p_i \cdot Y_j(i, t) dz_i - \int_{\mathcal{X}_i} p_i \cdot Y_j(i, t') dz_i \big)^2 dz_t dz_{t'} \Big) \Big] d\mathbf{x}_S$$

$$= \int_{\mathbf{x}_S \in \mathcal{X}_S} p(\mathbf{x}_S) \cdot \Big[ \sum_{j \in \mathcal{K}} \frac{1}{2} \Big( \int_{\mathcal{X}_t} \int_{\mathcal{X}_{t'}} p_t p_{t'} \big( \int_{\mathcal{X}_i} p_i \cdot (Y_j(i, t) - Y_j(i, t')) dz_i \big)^2 dz_t dz_{t'} \Big) \Big] d\mathbf{x}_S,$$

(49)

where

$$\int_{\mathcal{X}_t} \int_{\mathcal{X}_{t'}} p_t p_{t'} \big( \int_{\mathcal{X}_i} p_i \cdot (Y_j(i, t) - Y_j(i, t')) dz_i \big)^2 dz_t dz_{t'}$$

$$= \int_{\mathcal{X}_t} \int_{\mathcal{X}_{t'}} \int_{\mathcal{X}_i} \int_{\mathcal{X}_{i'}} p_t p_{t'} p_i p_{i'} \cdot (Y_j(i, t) - Y_j(i, t')) \cdot (Y_j(i', t) - Y_j(i', t')) dz_i dz_{i'} dz_t dz_{t'}.$$

Let $S' = S \cup \{i\}$, then we have:

$$\Delta_t(S', f)$$

$$= \int_{\mathbf{x}_{S'} \in \mathcal{X}_{S'}} p(\mathbf{x}_{S'}) \cdot \Big[ \sum_{j \in \mathcal{K}} \frac{1}{2} \Big( \int_{\mathcal{X}_t} \int_{\mathcal{X}_{t'}} p_t p_{t'} \cdot \big( Y_j(i, t) - Y_j(i, t') \big)^2 dz_t dz_{t'} \Big) \Big] d\mathbf{x}_{S'}$$

$$= \int_{\mathbf{x}_S \in \mathcal{X}_S} p(\mathbf{x}_S) \cdot \Big[ \sum_{j \in \mathcal{K}} \frac{1}{2} \int_{\mathcal{X}_i} p_i \cdot \Big( \int_{\mathcal{X}_t} \int_{\mathcal{X}_{t'}} p_t p_{t'} \cdot \big( Y_j(i, t) - Y_j(i, t') \big)^2 dz_t dz_{t'} \Big) dz_i \Big] d\mathbf{x}_S$$

(50)

$$= \int_{\mathbf{x}_S \in \mathcal{X}_S} p(\mathbf{x}_S) \cdot \Big[ \sum_{j \in \mathcal{K}} \frac{1}{2} \Big( \int_{\mathcal{X}_i} \int_{\mathcal{X}_t} \int_{\mathcal{X}_{t'}} p_i p_t p_{t'} \cdot \big( Y_j(i, t) - Y_j(i, t') \big)^2 dz_t dz_{t'} dz_i \Big) \Big] d\mathbf{x}_S,$$

where

$$\int_{\mathcal{X}_i} \int_{\mathcal{X}_t} \int_{\mathcal{X}_{t'}} p_i p_t p_{t'} \cdot \big( Y_j(i, t) - Y_j(i, t') \big)^2 dz_t dz_{t'} dz_i$$

$$= \int_{\mathcal{X}_i} \int_{\mathcal{X}_{i'}} \int_{\mathcal{X}_t} \int_{\mathcal{X}_{t'}} p_i p_{i'} p_t p_{t'} \cdot \big( Y_j(i, t) - Y_j(i, t') \big)^2 dz_t dz_{t'} dz_i dz_{i'}.$$

Let $\alpha(i, t, t')$ denote $Y_j(i, t) - Y_j(i, t')$ and $\beta(i', t, t')$ denote $Y_j(i', t) - Y_j(i', t')$, we can infer that

$$\Delta_t(S', f) - \Delta_t(S, f)$$

$$= \int_{\mathbf{x}_S \in \mathcal{X}_S} p(\mathbf{x}_S) \Big[ \sum_{j \in \mathcal{K}} \frac{1}{4} \Big( \int_{\mathcal{X}_t} \int_{\mathcal{X}_{t'}} \int_{\mathcal{X}_i} \int_{\mathcal{X}_{i'}} p_t p_{t'} p_i p_{i'} (\alpha(i, t, t') - \beta(i', t, t'))^2 dz_i dz_{i'} dz_t dz_{t'} \Big) \Big] d\mathbf{x}_S$$

$$\geq 0.$$

(51)

Consequently, the global sufficient value function $v_{\text{suff}}^g$ retains the supermodularity property for any valid classification function, regardless of whether its features are Boolean, discrete, continuous, or a mix thereof.

We now demonstrate that the local probabilistic sufficient value function is neither submodular nor supermodular. As a simple illustration, fix the uniform distribution on the three variables $\{x_1, x_2, x_3\}$ and define a Boolean function as follows:

$$f(x_1, x_2) = (x_1 \vee x_2) \wedge x_3. \tag{52}$$

Consider the input $\mathbf{x} = (1, 1, 1)$ which is classified as $1$. For the local probabilistic sufficient value function $v_{\text{suff}}^l$, we have that the following holds:

$$v_{\text{suff}}^l(\emptyset) = \frac{3}{8}, \quad v_{\text{suff}}^l(\{1\}) = \frac{1}{2}, \quad v_{\text{suff}}^l(\{2\}) = \frac{1}{2}, \quad v_{\text{suff}}^l(\{3\}) = \frac{3}{4},$$

$$v_{\text{suff}}^l(\{1,2\}) = \frac{1}{2}, \quad v_{\text{suff}}^l(\{1,3\}) = 1, \quad v_{\text{suff}}^l(\{2,3\}) = 1, \quad v_{\text{suff}}^l(\{1,2,3\}) = 1.$$

Since $\left(v_{\text{suff}}^l(\{1,2\}) - v_{\text{suff}}^l(\{2\})\right) < \left(v_{\text{suff}}^l(\{1\}) - v_{\text{suff}}^l(\emptyset)\right) < \left(v_{\text{suff}}^l(\{1,3\}) - v_{\text{suff}}^l(\{3\})\right)$, the function $v_{\text{suff}}^l$ is neither supermodular nor submodular.

$\square$

## F  PROOF OF PROPOSITION 4

**Proposition 4.** *While the local probabilistic contrastive setting (for any $\delta$) lacks submodularity — even when $\mathcal{D}$ is uniform, i.e., the value function $v_{con}^\ell$ is not submodular — in the global probabilistic setting (also for any $\delta$), when $\mathcal{D}$ exhibits feature independence, the value function $v_{con}^g$ is submodular.*

*Proof.* Once we establish that the global sufficient setting is supermodular, it directly follows that the global contrastive setting is submodular. Let $S \subseteq [n]$ be the set of features allowed to change, and suppose $S \subseteq S'$ with $i \notin S'$. To compare the marginal contributions $v_{\text{con}}^g(S \cup \{i\}) - v_{\text{con}}^g(S)$ and $v_{\text{con}}^g(S' \cup \{i\}) - v_{\text{con}}^g(S')$, we equivalently compare $\left(1 - v_{\text{suff}}^g(\bar{S} \setminus \{i\})\right) - \left(1 - v_{\text{suff}}^g(\bar{S})\right)$ and $\left(1 - v_{\text{suff}}^g(\bar{S}' \setminus \{i\})\right) - \left(1 - v_{\text{suff}}^g(\bar{S}')\right)$. Since $S \subseteq S'$, it follows that $\bar{S}' \subseteq \bar{S}$, and therefore we obtain:

$$\left(v_{\text{suff}}^g(\bar{S}) - v_{\text{suff}}^g(\bar{S} \setminus \{i\})\right) \geq \left(v_{\text{suff}}^g(\bar{S}') - v_{\text{suff}}^g(\bar{S}' \setminus \{i\})\right) \tag{53}$$

Which is equivalent to:

$$\left(v_{\text{con}}^g(S \cup \{i\}) - v_{\text{con}}^g(S)\right) \geq \left(v_{\text{con}}^g(S' \cup \{i\}) - v_{\text{con}}^g(S')\right). \tag{54}$$

Thus, the global contrastive value function $v_{\text{con}}^g$ is submodular, hence concluding this proof.

We will now demonstrate that the local probabilistic contrastive value function is neither submodular nor supermodular. As a simple illustration, fix the uniform distribution on the three variables $\{x_1, x_2, x_3\}$ and define a Boolean function as follows:

$$f(x_1, x_2) := (x_1 \vee x_2) \wedge x_3. \tag{55}$$

Consider the input $\mathbf{x} = (1, 1, 1)$ which is classified as $1$. For the local probabilistic contrastive value function $v_{\text{con}}^l$, we have:

$$v_{\text{con}}^l(\emptyset) = 1, \quad v_{\text{con}}^l(\{1\}) = 1, \quad v_{\text{con}}^l(\{2\}) = 1, \quad v_{\text{con}}^l(\{3\}) = \frac{1}{2},$$

$$v_{\text{con}}^l(\{1,2\}) = \frac{3}{4}, \quad v_{\text{con}}^l(\{1,3\}) = \frac{1}{2}, \quad v_{\text{con}}^l(\{2,3\}) = \frac{1}{2}, \quad v_{\text{con}}^l(\{1,2,3\}) = \frac{3}{8}.$$

Since $\left(v_{\text{con}}^l(\{1,2\}) - v_{\text{con}}^l(\{2\})\right) < \left(v_{\text{con}}^l(\{1,2,3\}) - v_{\text{con}}^l(\{2,3\})\right) < \left(v_{\text{con}}^l(\{1\}) - v_{\text{con}}^l(\emptyset)\right)$, the function $v_{\text{con}}^l$ is neither submodular nor supermodular.

$\square$

In the rest of this section, we show that the feature independence assumption is not just sufficient but also necessary for supermodularity. We illustrate this with the following counterexample.

Define a Boolean function:

$$f(x_1, x_2) := x_1 \tag{56}$$

For this boolean function, it holds that $f(00) = 0$, $f(01) = 0$, $f(10) = 1$, and $f(11) = 1$. Moreover, define a distribution $\mathcal{D}$ on $x_1$ and $x_2$. Recall that the global sufficient value function is:

$$\mathbb{E}_{\mathbf{x}\sim\mathcal{D}}[\mathbf{Pr}_{\mathbf{z}\sim\mathcal{D}}(f(\mathbf{z}) = f(\mathbf{x}) \mid \mathbf{z}_S = \mathbf{x}_S)] = \sum_{\mathbf{x}\sim\mathcal{D}} \mathbf{Pr}(\mathbf{x}) \cdot \mathbf{Pr}_{\mathbf{z}\sim\mathcal{D}}(f(\mathbf{z}) = f(\mathbf{x}) \mid \mathbf{z}_S = \mathbf{x}_S)$$

In addition, if we can show that

$$\Delta_t(S, f) \leq \Delta_t(S', f),$$

where $S \subseteq S'$ and $t \notin S'$, then the global sufficient value function is supermodular. When $S = \emptyset$, we have that the following holds:

$$\mathbb{E}_{\mathbf{x}\sim\mathcal{D}}[\mathbf{Pr}_{\mathbf{z}\sim\mathcal{D}}(f(\mathbf{z}) = f(\mathbf{x}) \mid \mathbf{z}_S = \mathbf{x}_S)]$$
$$= \mathbf{Pr}(x_1 = 0, x_2 = 0) \cdot \mathbf{Pr}_{\mathbf{z}\sim\mathcal{D}}(f(\mathbf{z}) = 0) + \mathbf{Pr}(x_1 = 0, x_2 = 1) \cdot \mathbf{Pr}_{\mathbf{z}\sim\mathcal{D}}(f(\mathbf{z}) = 0) +$$
$$\mathbf{Pr}(x_1 = 1, x_2 = 0) \cdot \mathbf{Pr}_{\mathbf{z}\sim\mathcal{D}}(f(\mathbf{z}) = 1) + \mathbf{Pr}(x_1 = 1, x_2 = 1) \cdot \mathbf{Pr}_{\mathbf{z}\sim\mathcal{D}}(f(\mathbf{z}) = 1)$$

Which is also equivalent to:

$$\mathbf{Pr}(x_1 = 0, x_2 = 0) \cdot \big(\mathbf{Pr}(x_1 = 0, x_2 = 0) + \mathbf{Pr}(x_1 = 0, x_2 = 1)\big) +$$
$$\mathbf{Pr}(x_1 = 0, x_2 = 1) \cdot \big(\mathbf{Pr}(x_1 = 0, x_2 = 0) + \mathbf{Pr}(x_1 = 0, x_2 = 1)\big) +$$
$$\mathbf{Pr}(x_1 = 1, x_2 = 0) \cdot \big(\mathbf{Pr}(x_1 = 1, x_2 = 0) + \mathbf{Pr}(x_1 = 1, x_2 = 1)\big) +$$
$$\mathbf{Pr}(x_1 = 1, x_2 = 1) \cdot \big(\mathbf{Pr}(x_1 = 1, x_2 = 0) + \mathbf{Pr}(x_1 = 1, x_2 = 1)\big)$$
$$= \big(\mathbf{Pr}(x_1 = 0, x_2 = 0) + \mathbf{Pr}(x_1 = 0, x_2 = 1)\big)^2 + \big(\mathbf{Pr}(x_1 = 1, x_2 = 0) + \mathbf{Pr}(x_1 = 1, x_2 = 1)\big)^2$$

When $S = \{2\}$, we have that the following condition holds:

$$\mathbb{E}_{\mathbf{x}\sim\mathcal{D}}[\mathbf{Pr}_{\mathbf{z}\sim\mathcal{D}}(f(\mathbf{z}) = f(\mathbf{x}) \mid \mathbf{z}_S = \mathbf{x}_S)]$$
$$= \mathbf{Pr}(x_1 = 0, x_2 = 0) \cdot \mathbf{Pr}_{\mathbf{z}\sim\mathcal{D}}(f(\mathbf{z}) = 0 \mid x_2 = 0) +$$
$$\mathbf{Pr}(x_1 = 0, x_2 = 1) \cdot \mathbf{Pr}_{\mathbf{z}\sim\mathcal{D}}(f(\mathbf{z}) = 0 \mid x_2 = 1) +$$
$$\mathbf{Pr}(x_1 = 1, x_2 = 0) \cdot \mathbf{Pr}_{\mathbf{z}\sim\mathcal{D}}(f(\mathbf{z}) = 1 \mid x_2 = 0) +$$
$$\mathbf{Pr}(x_1 = 1, x_2 = 1) \cdot \mathbf{Pr}_{\mathbf{z}\sim\mathcal{D}}(f(\mathbf{z}) = 1 \mid x_2 = 1)$$

This can also be rearranged as follows:

$$\mathbf{Pr}(x_1 = 0, x_2 = 0) \cdot \frac{\mathbf{Pr}(x_1 = 0, x_2 = 0)}{\mathbf{Pr}(x_1 = 0, x_2 = 0) + \mathbf{Pr}(x_1 = 1, x_2 = 0)} +$$
$$\mathbf{Pr}(x_1 = 0, x_2 = 1) \cdot \frac{\mathbf{Pr}(x_1 = 0, x_2 = 1)}{\mathbf{Pr}(x_1 = 0, x_2 = 1) + \mathbf{Pr}(x_1 = 1, x_2 = 1)} +$$
$$\mathbf{Pr}(x_1 = 1, x_2 = 0) \cdot \frac{\mathbf{Pr}(x_1 = 1, x_2 = 0)}{\mathbf{Pr}(x_1 = 0, x_2 = 0) + \mathbf{Pr}(x_1 = 1, x_2 = 0)} +$$
$$\mathbf{Pr}(x_1 = 1, x_2 = 1) \cdot \frac{\mathbf{Pr}(x_1 = 1, x_2 = 1)}{\mathbf{Pr}(x_1 = 0, x_2 = 1) + \mathbf{Pr}(x_1 = 1, x_2 = 1)}$$
$$= \frac{\mathbf{Pr}(x_1 = 0, x_2 = 0)^2 + \mathbf{Pr}(x_1 = 1, x_2 = 0)^2}{\mathbf{Pr}(x_1 = 0, x_2 = 0) + \mathbf{Pr}(x_1 = 1, x_2 = 0)} + \frac{\mathbf{Pr}(x_1 = 0, x_2 = 1)^2 + \mathbf{Pr}(x_1 = 1, x_2 = 1)^2}{\mathbf{Pr}(x_1 = 0, x_2 = 1) + \mathbf{Pr}(x_1 = 1, x_2 = 1)}$$

In contrast, when $S = \{1\}$, we have the the following holds:

$$\mathbb{E}_{\mathbf{x}\sim\mathcal{D}}[\mathbf{Pr}_{\mathbf{z}\sim\mathcal{D}}(f(\mathbf{z}) = f(\mathbf{x}) \mid \mathbf{z}_S = \mathbf{x}_S)]$$
$$= \mathbf{Pr}(x_1 = 0, x_2 = 0) \cdot \mathbf{Pr}_{\mathbf{z}\sim\mathcal{D}}(f(\mathbf{z}) = 0 \mid x_1 = 0) +$$
$$\mathbf{Pr}(x_1 = 0, x_2 = 1) \cdot \mathbf{Pr}_{\mathbf{z}\sim\mathcal{D}}(f(\mathbf{z}) = 0 \mid x_1 = 0) +$$
$$\mathbf{Pr}(x_1 = 1, x_2 = 0) \cdot \mathbf{Pr}_{\mathbf{z}\sim\mathcal{D}}(f(\mathbf{z}) = 1 \mid x_1 = 1) +$$
$$\mathbf{Pr}(x_1 = 1, x_2 = 1) \cdot \mathbf{Pr}_{\mathbf{z}\sim\mathcal{D}}(f(\mathbf{z}) = 1 \mid x_1 = 1)$$

This can similarly be rearranged as follows:

$$\mathbf{Pr}(x_1 = 0, x_2 = 0) \cdot \frac{\mathbf{Pr}(x_1 = 0, x_2 = 0) + \mathbf{Pr}(x_1 = 0, x_2 = 1)}{\mathbf{Pr}(x_1 = 0, x_2 = 0) + \mathbf{Pr}(x_1 = 0, x_2 = 1)} +$$

$$\mathbf{Pr}(x_1 = 0, x_2 = 1) \cdot \frac{\mathbf{Pr}(x_1 = 0, x_2 = 0) + \mathbf{Pr}(x_1 = 0, x_2 = 1)}{\mathbf{Pr}(x_1 = 0, x_2 = 0) + \mathbf{Pr}(x_1 = 0, x_2 = 1)} +$$

$$\mathbf{Pr}(x_1 = 1, x_2 = 0) \cdot \frac{\mathbf{Pr}(x_1 = 1, x_2 = 0) + \mathbf{Pr}(x_1 = 1, x_2 = 1)}{\mathbf{Pr}(x_1 = 1, x_2 = 0) + \mathbf{Pr}(x_1 = 1, x_2 = 1)} +$$

$$\mathbf{Pr}(x_1 = 1, x_2 = 1) \cdot \frac{\mathbf{Pr}(x_1 = 1, x_2 = 0) + \mathbf{Pr}(x_1 = 1, x_2 = 1)}{\mathbf{Pr}(x_1 = 1, x_2 = 0) + \mathbf{Pr}(x_1 = 1, x_2 = 1)} = 1$$

Finally, when $S = \{1, 2\}$, we have the following condition:

$$\mathbb{E}_{\mathbf{x} \sim \mathcal{D}}[\mathbf{Pr}_{\mathbf{z} \sim \mathcal{D}}(f(\mathbf{z}) = f(\mathbf{x}) \mid \mathbf{z}_S = \mathbf{x}_S)]$$
$$= \mathbf{Pr}(x_1 = 0, x_2 = 0) \cdot \mathbf{Pr}_{\mathbf{z} \sim \mathcal{D}}(f(\mathbf{z}) = 0 \mid x_1 = 0, x_2 = 0) +$$
$$\mathbf{Pr}(x_1 = 0, x_2 = 1) \cdot \mathbf{Pr}_{\mathbf{z} \sim \mathcal{D}}(f(\mathbf{z}) = 0 \mid x_1 = 0, x_2 = 1) +$$
$$\mathbf{Pr}(x_1 = 1, x_2 = 0) \cdot \mathbf{Pr}_{\mathbf{z} \sim \mathcal{D}}(f(\mathbf{z}) = 1 \mid x_1 = 1, x_2 = 0) +$$
$$\mathbf{Pr}(x_1 = 1, x_2 = 1) \cdot \mathbf{Pr}_{\mathbf{z} \sim \mathcal{D}}(f(\mathbf{z}) = 1 \mid x_1 = 1, x_2 = 1)$$

This can likewise be reordered as follows:

$$\mathbf{Pr}(x_1 = 0, x_2 = 0) \cdot \frac{\mathbf{Pr}(x_1 = 0, x_2 = 0)}{\mathbf{Pr}(x_1 = 0, x_2 = 0)} + \mathbf{Pr}(x_1 = 0, x_2 = 1) \cdot \frac{\mathbf{Pr}(x_1 = 0, x_2 = 1)}{\mathbf{Pr}(x_1 = 0, x_2 = 1)} +$$

$$\mathbf{Pr}(x_1 = 1, x_2 = 0) \cdot \frac{\mathbf{Pr}(x_1 = 1, x_2 = 0)}{\mathbf{Pr}(x_1 = 1, x_2 = 0)} + \mathbf{Pr}(x_1 = 1, x_2 = 1) \cdot \frac{\mathbf{Pr}(x_1 = 1, x_2 = 1)}{\mathbf{Pr}(x_1 = 1, x_2 = 1)}$$

$$= 1$$

Let $t = 2$. Evidently, we have $\Delta_t(\{1\}, f) = 0$. Next, let us analyse $\Delta_t(\emptyset, f)$. We note that if we assume feature independence, we observe that:

$$\left(\mathbf{Pr}(x_1 = 0, x_2 = 0) + \mathbf{Pr}(x_1 = 0, x_2 = 1)\right)^2 + \left(\mathbf{Pr}(x_1 = 1, x_2 = 0) + \mathbf{Pr}(x_1 = 1, x_2 = 1)\right)^2$$
$$= \mathbf{Pr}(x_1 = 0)^2 + \mathbf{Pr}(x_1 = 1)^2,$$

Additionally, another condition that holds is:

$$\frac{\mathbf{Pr}(x_1 = 0, x_2 = 0)^2 + \mathbf{Pr}(x_1 = 1, x_2 = 0)^2}{\mathbf{Pr}(x_1 = 0, x_2 = 0) + \mathbf{Pr}(x_1 = 1, x_2 = 0)} + \frac{\mathbf{Pr}(x_1 = 0, x_2 = 1)^2 + \mathbf{Pr}(x_1 = 1, x_2 = 1)^2}{\mathbf{Pr}(x_1 = 0, x_2 = 1) + \mathbf{Pr}(x_1 = 1, x_2 = 1)}$$
$$= \mathbf{Pr}(x_1 = 0)^2 + \mathbf{Pr}(x_1 = 1)^2.$$

This implies $\Delta_t(\emptyset, f) = 0$, that is, $\Delta_t(\emptyset, f) \leq \Delta_t(\{1\}, f)$. However, without the assumption of feature independence, we cannot guarantee $\Delta_t(\emptyset, f) = 0$. For example, let:

$$\mathbf{Pr}(x_1 = 0, x_2 = 0) = P_{00}, \quad \mathbf{Pr}(x_1 = 0, x_2 = 1) = P_{01}$$

and

$$\mathbf{Pr}(x_1 = 1, x_2 = 0) = P_{10}, \quad \mathbf{Pr}(x_1 = 1, x_2 = 1) = P_{11}$$

Then, by simplifying $\Delta_t(\emptyset, f)$ we get that the following holds:

$$\Delta_t(\emptyset, f) = \left[\frac{P_{00}^2 + P_{10}^2}{P_{00} + P_{10}} + \frac{P_{01}^2 + P_{11}^2}{P_{01} + P_{11}}\right] - \left[\left(P_{00} + P_{01}\right)^2 + \left(P_{10} + P_{11}\right)^2\right] \tag{57}$$

We then let $P_{00} = 0.4$, $P_{01} = 0.2$, $P_{10} = 0.1$, and $P_{11} = 0.3$, and get that the following holds:

$$\Delta_t(\emptyset, f) = \left[\frac{(0.4)^2 + (0.1)^2}{0.5} + \frac{(0.2)^2 + (0.3)^2}{0.5}\right] - \left[\left(0.4 + 0.2\right)^2 + \left(0.1 + 0.3\right)^2\right] = 0.08. \tag{58}$$

This implies that $\Delta_t(\emptyset, f) > \Delta_t(1, f)$, indicating that — unlike in the monotonicity setting where it was not required — feature independence is a necessary condition for supermodularity.

## G    PROOF OF PROPOSITION 5

**Proposition 5.** *Computing Algorithm 1 with the* local *value functions $v_{con}^{\ell}$ and $v_{suff}^{\ell}$ does not always converge to a subset minimal $\delta$-sufficient/contrastive reason. However, computing it with the* global *value functions $v_{con}^{g}$ or $v_{suff}^{g}$ necessarily produces subset minimal $\delta$-sufficient/contrastive reasons.*

*Proof.* We note that this proposition follows directly from our proof that both $v_{suff}^{g}$ and $v_{con}^{g}$ are monotone non-decreasing (as established in Proposition 2), along with other relevant conclusions drawn in prior work. In the remainder of this section, we elaborate on how this monotonicity property leads to the stated corollary.

The convergence of Algorithm 1 to a subset-minimal $\delta$ global sufficient or contrastive reason with respect to the value functions $v_{con}^{g}$ and $v^{g}$suff follows directly from the monotonicity property, for which prior work (Ignatiev et al., 2019; Wu et al., 2024b; Arenas et al., 2022) showed that this type of greedy algorithm yields a subset-minimal explanation when the underlying value function is monotone non-decreasing. The algorithm halts when, for all $i \in S$, the value $v(S \setminus \{i\})$ drops below $\delta$, ensuring that although $v(S) \geq \delta$ (a maintained invariant), removing any element causes the condition to fail. Due to monotonicity, this implies that no proper subset of $S$ satisfies the condition, guaranteeing that $S$ is subset-minimal.

We now turn to explain why Algorithm 1 does not converge to a subset-minimal solution in the local setting. We note that it is well known that the local probabilistic sufficient value function (and likewise the local probabilistic contrastive value function) is not monotone Arenas et al. (2022); Izza et al. (2023); Subercaseaux et al. (2025); Izza et al. (2024b). As a simple illustration, fix the uniform distribution on the two binary variables $\{x_1, x_2\}$ and define a Boolean function as follows:

$$f(x_1, x_2) := x_1 \vee x_2. \tag{59}$$

Consider the input $\mathbf{x} = (0, 1)$ which is classified as 1, then:

$$v_{\text{suff}}^{\ell}(\emptyset) = \frac{3}{4}, \;\; v_{\text{suff}}^{\ell}(\{1\}) = \frac{1}{2}, \;\; v_{\text{suff}}^{\ell}(\{2\}) = 1, \;\; v_{\text{suff}}^{\ell}(\{1, 2\}) = 1. \tag{60}$$

Because $v_{\text{suff}}^{\ell}(\emptyset) > v_{\text{suff}}^{\ell}(\{1\}) < v_{\text{suff}}^{\ell}(\{1, 2\})$, the function $v_{\text{suff}}^{\ell}$ is not monotone. Likewise, it can be computed that:

$$v_{\text{con}}^{\ell}(\emptyset) = 1, \;\; v_{\text{con}}^{\ell}(\{1\}) = 1, \;\; v_{\text{con}}^{\ell}(\{2\}) = \frac{1}{2}, \;\; v_{\text{con}}^{\ell}(\{1, 2\}) = \frac{3}{4}. \tag{61}$$

(For $v_{\text{con}}^{\ell}$, the $S$ denotes the set of features that are allowed to vary.) Because $v_{\text{con}}^{\ell}(\emptyset) > v_{\text{con}}^{\ell}(\{2\}) < v_{\text{con}}^{\ell}(\{1, 2\})$, the function $v_{\text{con}}^{\ell}$ is also non-monotone. $\square$

## H    PROOF OF THEOREM 1

**Theorem 1.** *If $f$ is a decision tree and the probability term $v_{suff}^{g}$ can be computed in polynomial time given the distribution $\mathcal{D}$ (which holds for independent distributions, among others), then obtaining a subset-minimal* global *$\delta$-sufficient reason can be obtained in* polynomial time. *However, unless PTIME=NP, no polynomial-time* algorithm exists for computing a *local $\delta$-sufficient reason for decision trees even under independent distributions.*

*Proof.* We begin by noting that for decision trees, assuming that $\mathcal{D}$ represents independent distributions, the computation of the value function:

$$v_{\text{suff}}^{g}(S) = \mathbb{E}_{\mathbf{x} \sim \mathcal{D}}[\mathbf{Pr}_{\mathbf{z} \sim \mathcal{D}}(f(\mathbf{x}) = f(\mathbf{z}) \mid \mathbf{x}_S = \mathbf{z}_S)] \tag{62}$$

Can be carried out in polynomial time using the following procedure. Thanks to the tractability of decision trees, we can iterate over all pairs of leaf nodes, each corresponding to partial assignments

$\mathbf{x}_S$ and $\mathbf{z}_{S'}$. For each such pair, if both leaf nodes yield the same prediction under $f$, we compute the corresponding term in the expected value $\mathbb{E}_{\mathbf{x}\sim\mathcal{D}}[\mathbf{Pr}_{\mathbf{z}\sim\mathcal{D}}(f(\mathbf{x}) = f(\mathbf{z}) \mid \mathbf{x}_S = \mathbf{z}_S)]$ by multiplying the respective feature-wise probabilities over the shared features of the two vectors. Under the feature independence assumption, these probabilities decompose, allowing the full expectation to be computed by summing the contributions of all such matching leaf pairs.

Since all probabilities involved are provided as part of the input and each step involves only polynomial-time operations, the entire procedure runs in polynomial time. This establishes that every step in Algorithm 1 is efficient, and thus the algorithm as a whole runs in polynomial time. Finally, combining this with Lemma 2, which proves that $v_{\text{suff}}^g$ is monotone non-decreasing, we conclude that Algorithm 1 converges to a subset-minimal explanation. This is because the algorithm halts when, for all $i \in S$, the value $v(S \setminus \{i\})$ falls below $\delta$, while $v(S) \geq \delta$ is preserved throughout. By monotonicity, this guarantees that no strict subset of $S$ satisfies the condition, ensuring $S$ is indeed subset-minimal.

For the remaining part of the claim, the result follows from (Arenas et al., 2022), which showed that, assuming P$\neq$NP, there is no polynomial-time algorithm for computing a subset-minimal $\delta$-sufficient reason for decision trees. While this was proven specifically under the uniform distribution, the hardness clearly extends to independent distributions, which include this case. Combined, these results establish the complexity separation between the local and global variants, thus concluding our proof.

$\square$

## I    PROOF OF PROPOSITION 6

**Proposition 6.** *For any model $f$, and empirical distribution $\mathcal{D}$ — computing a subset-minimal* global *$\delta$-sufficient or $\delta$-contrastive reason for $f$ can be done in polynomial time.*

*Proof.* The computation of the global sufficient probability function:

$$v_{\text{suff}}^g(S) = \mathbb{E}_{\mathbf{x}\sim\mathcal{D}}[\mathbf{Pr}_{\mathbf{z}\sim\mathcal{D}}(f(\mathbf{x}) = f(\mathbf{z}) \mid \mathbf{x}_S = \mathbf{z}_S)] \tag{63}$$

or the computation of the global contrastive probability function

$$v_{\text{con}}^g(S) = \mathbb{E}_{\mathbf{x}\sim\mathcal{D}}[\mathbf{Pr}_{\mathbf{z}\sim\mathcal{D}}(f(\mathbf{x}) \neq f(\mathbf{z}) \mid \mathbf{x}_{\bar{S}} = \mathbf{z}_{\bar{S}})] \tag{64}$$

Can be performed in polynomial time when $\mathcal{D}$ is selected from the class of empirical distributions. This is achieved by iterating over pairs of instances $\mathbf{x}, \mathbf{z}$ within the dataset and running an inference through $f(\mathbf{x})$ and $f(\mathbf{z})$ to compute the expected values at each step. Consequently, determining whether the probability function exceeds or falls below a given threshold $\delta$ can also be accomplished in polynomial time. Furthermore, leveraging the proof of non-decreasing monotonicity (Proposition D) of the global probability function, we can utilize algorithm 1 for contrastive reason search or for sufficient reason search. By executing a linear number of polynomial queries, a subset-minimal explanation (whether a subset minimal contrastive reason or a subset minimal sufficient reason) can thus be obtained in polynomial time.

$\square$

## J    PROOF OF THEOREM 2

**Theorem 2.** *Assuming $f$ is a neural network or a tree ensemble, and $\mathcal{D}$ is an empirical distribution — there exist* polynomial-time *algorithms for obtaining subset minimal* global *$\delta$-sufficient and contrastive reasons. However, unless PTIME=NP, there is* no polynomial time *algorithm for computing a subset minimal* local *$\delta$-sufficient reason or a subset minimal* local *$\delta$-contrastive reason.*

*Proof.* The first part of the proof follows directly from Proposition 6, which established that this holds for any model, since both $v_{\text{suff}}^g(S) = \mathbb{E}_{\mathbf{x}\sim\mathcal{D}}[\mathbf{Pr}_{\mathbf{z}\sim\mathcal{D}}(f(\mathbf{x}) = f(\mathbf{z}) \mid \mathbf{x}_S = \mathbf{z}_S)]$ and $v_{\text{con}}^g(S) = \mathbb{E}_{\mathbf{x}\sim\mathcal{D}}[\mathbf{Pr}_{\mathbf{z}\sim\mathcal{D}}(f(\mathbf{x}) \neq f(\mathbf{z}) \mid \mathbf{x}_{\bar{S}} = \mathbf{z}_{\bar{S}})]$ can be computed in polynomial time from

the empirical distribution. For the second part, we proceed by proving two lemmas: one establishing the intractability of computing the sufficient case, and the other addressing the contrastive case.

**Lemma 6.** *Unless PTIME= NP, then there is no polynomial time algorithm for computing a subset minimal local $\delta$-sufficient reason for either a tree ensemble or a neural network, under empirical distributions.*

*Proof.* We will first prove this claim for neural networks and then extend the result to tree ensembles. We will establish this claim by demonstrating that if a polynomial-time algorithm exists for computing a subset-minimal local $\delta$ sufficient reason for neural networks, then it would enable us to solve the classic NP-complete *CNF-SAT* problem in polynomial time. The CNF-SAT problem is defined as follows:

---

**CNF-SAT**:
**Input**: A formula in conjunctive normal form (CNF): $\phi$.
**Output**: *Yes*, if there exists an assignment to the $n$ literals of $\phi$ such that $\phi$ is evaluated to True, and *No* otherwise

---

Our proof will also utilize the following lemma (with its proof provided in Barceló et al. (2020)):

**Lemma 7.** *Any boolean circuit $\phi$ can be encoded into an equivalent MLP over the binary domain $\{0,1\}^n \to \{0,1\}$ in polynomial time.*

*Proof.* We will actually establish hardness for a simpler, more specific case, which will consequently imply hardness for the more general setting. In this case, we assume that the empirical distribution $\mathcal{D}$ consists of a single element, which, for simplicity, we take to be the zero vector $\mathbf{0}_n$. Since the sufficiency of $S$ in this scenario depends only on the local instance $\mathbf{x}$ and the single instance $\mathbf{0}_n$ in our empirical distribution, the nature of the input format (whether discrete, continuous, etc.) does not affect the result. Therefore, the hardness results hold universally across all these settings.

Similarly to $\mathbf{0}_n$, we define $\mathbf{1}_n$ as an $n$-dimensional vector consisting entirely of ones. Given an input $\phi$, we initially assign $\mathbf{1}_n$ to $\phi$, effectively setting all variables to True. If $\phi$ evaluates to True under this assignment, then a satisfying truth assignment exists. Therefore, we assume that $\phi(\mathbf{1}_n) = 0$. We now introduce the following definitions:

$$\phi_2 := (x_1 \wedge x_2 \wedge \ldots x_n),$$
$$\phi' := \phi \vee \phi_2 \tag{65}$$

$\phi'$, while no longer a CNF, can still be transformed into an MLP $f$ using Lemma 8, ensuring that $f$ behaves equivalently to $\phi'$ over the domain $\{0,1\}^n$. Given our assumption that computing a subset-minimal local $\delta$-sufficient reason is feasible in polynomial time, we can determine one for the instance $\langle f, \mathbf{x} := \mathbf{1}_n \rangle$, $\delta := 1$, noting that the empirical distribution $\mathcal{D}$ is simply defined over the single data point $\mathbf{0}_n$.

We now assert that the subset-minimal $\delta$-sufficient reason generated for $\langle f, \mathbf{x} \rangle$ encompasses the entire input space, i.e., $S = \{1, \ldots, n\}$, if and only if $\langle \phi \rangle \notin$CNF-SAT.

Let us assume that $S = \{1, \ldots, n\}$. Since $S$ is a $\delta$-sufficient reason for $\langle f, \mathbf{x} \rangle$, this simply means that setting the complementary set to any value maintains the prediction. Since the complementary set is $\emptyset$ in this case, this trivially holds. The fact that $S$ is subset-minimal means that any other subset $S' \subseteq S$ satisfies $v(S') < \delta = 1$. Since the probability function $\mathbf{Pr}_{\mathbf{z} \sim \mathcal{D}}(f(\mathbf{x}) = f(\mathbf{z}) \mid \mathbf{x}_S = \mathbf{z}_S)$ is determined by a single point (the distribution contains only the point $\mathbf{0}_n$), the probability function can only take values of 1 or 0. Hence, we also know that $v(S') = 0$. This tells us that, aside from the subset $S = \{1, \ldots, n\}$, for any subset $S' \subseteq S$, fixing the features in $S'$ to 1 and the rest to 0 does not result in a classification outcome of 1. Since the $\phi_2$ component within $\phi'$ is True only if all features are assigned 1, this directly implies that $\phi$ is assigned False for any of these inputs. Since we already know that $\phi$ does not return a True answer for the vector assignment $\mathbf{1}_n$ (as verified at the beginning), and now we have established that the same holds for all other input vectors, we conclude that $\phi \notin$ *CNF-SAT*.

Now, suppose that $S$ is a subset that is strictly contained within $\{1, \ldots, n\}$. Given that $S$ is sufficient under our definitions of the distribution $\mathcal{D}$ with $\delta = 1$, we can apply the same reasoning as before to

conclude that $v(S) = 1$. This implies that setting the features in $S$ to 1 while setting the remaining features to 0 ensures that the function $f$ evaluates to 1. Since this assignment is necessarily not a vector consisting entirely of ones, it follows that the $\phi_2$ component within $\phi'$ must be False. Consequently, the $\phi$ component must be True, which implies that $\langle\phi\rangle \in$ CNF-SAT. This completes the proof.

We have established that the following claim holds for neural networks. However, extending the proof to tree ensembles requires a minor and straightforward adaptation. To do so, we will utilize the following lemma, which has been noted in several previous works (Ordyniak et al., 2024; Audemard et al., 2022b):

**Lemma 8.** *Any CNF or DNF $\phi$ can be encoded into an equivalent random forest classifier over the binary domain $\{0, 1\}^n \rightarrow \{0, 1\}$ in polynomial time.*

*Proof.* We observe that we can apply the same process used in our proof for neural networks, where we encoded $\phi'$ into an equivalent neural network. However, $\phi'$ is no longer a valid CNF due to our construction (though encoding it into an MLP was not an issue, as any Boolean circuit can be transformed into an MLP). Nevertheless, since $\phi'$ consists of a conjunction of only two terms, we can easily represent it as an equivalent CNF:

$$\phi' := (c_1 \vee \phi_2) \wedge (c_2 \vee \phi_2) \wedge \ldots (c_m \vee \phi_2) \tag{66}$$

Where each $c_i$ is a disjunction of a few terms. Consequently, $\phi'$ is a valid CNF, allowing us to transform it into an equivalent random forest classifier. The reduction we outlined for MLPs applies directly to these models as well, thereby completing the proof for both model families.

□

**Lemma 9.** *Unless PTIME= NP, then there is no polynomial time algorithm for computing a subset minimal local $\delta$-contrastive reason for either a tree ensemble or a neural network, under empirical distributions.*

*Proof.* We will present a proof analogous to the one in Lemma 6. Specifically, we will once again utilize the classical NP-hard CNF-SAT problem defined in Lemma 6. In particular, given a Boolean formula $\phi$, we will demonstrate that determining a subset-minimal contrastive reason — whether for a neural network or a tree ensemble — allows us to decide the satisfiability of $\phi$.

First, we check whether assigning all variables in $\phi$ to 1 evaluates $\phi$ to True. If so, the formula is satisfiable, and we have determined its satisfiability. Otherwise, we use Lemma 8 to encode the CNF formula as a neural network $f$. Next, we compute a subset-minimal $\delta$-sufficient reason by setting $\mathcal{D}$ as the empirical distribution containing only a single data point $\mathbf{0}_n$, following a similar procedure to Lemma 6. Additionally, we set $\delta = 1$ and compute a subset-minimal $\delta$-contrastive reason concerning $\langle f, \mathbf{x}\rangle$.

We will now demonstrate that if any subset-minimal $\delta$ contrastive reason obtained for $\phi$ is satisfiable, then $\phi$ itself is satisfiable. Conversely, if no subset-minimal $\delta$ contrastive reason is obtained, then $\phi$ is unsatisfiable. The validity of this claim follows a reasoning similar to that provided in Lemma 6. Specifically, the term $\mathbf{Pr}_{\mathbf{z}\sim\mathcal{D}}(f(\mathbf{x}) \neq f(\mathbf{z}) \mid \mathbf{x}_{\bar{S}} = \mathbf{z}_{\bar{S}})$, where the distribution $\mathcal{D}$ considers sampling from a single datapoint, can set the probability to either 0 or 1. Furthermore, since we are searching for a $\delta = 1$ contrastive reason, this is equivalent to asking whether there exists an assignment that changes the classification of $f$, which corresponds to modifying the assignment of $\phi$ from False to True. If such an assignment exists, then $\phi$ is satisfiable. However, if no subset-minimal contrastive reason exists, then no subset of features fixed to zero — when the complementary set is set to ones — evaluates to true. This is equivalent to stating that no assignment evaluates $f$ (and consequently $\phi$) to True, implying that $\phi$ is unsatisfiable.

To extend the proof from neural networks to tree ensembles, we can follow the same procedure outlined in Lemma 6, encoding the CNF formula into an equivalent random forest classifier. Consequently, the proof remains valid for tree ensembles, thereby concluding the proof.

□

## K    PROOF OF THEOREM 3

**Theorem 3.** *Given a neural network or tree ensemble $f$ and an empirical distribution $\mathcal{D}$ over a fixed dataset D, Algorithm 2 yields a constant $\mathcal{O}\left(\ln\left(\frac{v_{con}^g([n])}{\min_{i\in[n]} v_{con}^g(\{i\})}\right)\right)$-approximation, bounded by $\mathcal{O}(\ln(|D|))$, for computing a global cardinally minimal $\delta$-contrastive reason for $f$, assuming feature independence. In contrast, unless PTIME=NP, no bounded approximation exists for computing a local cardinally minimal $\delta$-contrastive reason for any $\langle f, \boldsymbol{x}\rangle$, even when $|D| = 1$.*

*Proof.* We divide the proof of the theorem into two lemmas, covering both the approximation guarantee and the result on the absence of a bounded approximation.

**Lemma 10.** *Given a neural network or tree ensemble $f$ and an empirical distribution $\mathcal{D}$ over a fixed dataset D, Algorithm 2 yields a constant $\mathcal{O}\left(\ln\left(\frac{v_{con}^g([n])}{\min_{i\in[n]} v_{con}^g(\{i\})}\right)\right)$-approximation, bounded by $\mathcal{O}(\ln(|D|))$, for computing a global cardinally minimal $\delta$-contrastive reason for $f$, assuming feature independence.*

*Proof.* We will in fact prove a stronger claim, showing that this holds for *any* model, provided the trivial condition that its inference time is computable in polynomial time, along with one additional mild condition that we will detail later — both of which apply to both our neural network and tree ensemble formalizations.

We begin by noting that, since we are working with empirical distributions, the computation of the global contrastive probability value function:

$$v_{\mathrm{con}}^g(S) = \mathbb{E}_{\mathbf{x}\sim\mathcal{D}}[\mathbf{Pr}_{\mathbf{z}\sim\mathcal{D}}(f(\mathbf{x}) \neq f(\mathbf{z}) \mid \mathbf{x}_{\bar{S}} = \mathbf{z}_{\bar{S}})] \tag{67}$$

Can be computed in polynomial time by iterating over all pairs $\mathbf{x}, \mathbf{z}$ in the dataset D, as previously established in Proposition 2. Since Algorithm 2 performs only a linear number of these polynomial-time queries, its total runtime is therefore polynomial.

Regarding the approximation, the classical work by Wolsey et al. (Wolsey, 1982) established a harmonic-series-based approximation guarantee for monotone non-decreasing submodular functions $v$ with integer values. More generally, their result yields an approximation factor of $\mathcal{O}\left(\ln\left(\frac{v([n])}{\min_{i\in[n]} v(\{i\})}\right)\right)$. We showed in Proposition 2 that the value function $v_{\mathrm{con}}^g$ is monotone non-decreasing, and under feature independence, Proposition 4 establishes its submodularity. Combined with the fact that Algorithm 2 runs in polynomial time, this directly yields an approximation guarantee of $\mathcal{O}\left(\ln\left(\frac{v_{\mathrm{con}}^g([n])}{\min_{i\in[n]} v_{\mathrm{con}}^g(\{i\})}\right)\right)$.

However, we must also show that the expression $\mathcal{O}\left(\ln\left(\frac{v_{\mathrm{con}}^g([n])}{\min_{i\in[n]} v_{\mathrm{con}}^g(\{i\})}\right)\right)$ is both finite and bounded by $\mathcal{O}(\ln(|D|))$, implying that it is effectively constant, due to the assumption of a fixed dataset $|D|$. To ensure this, we begin by confirming that the expression is well-defined and finite. We then proceed to establish the desired bounds. To do so, we introduce a preprocessing step in which we eliminate "redundant" elements—those that could theoretically cause the denominator $\min_{i\in[n]} v_{\mathrm{con}}^g(\{i\})$ to be zero. We begin by formally defining what we mean by redundancy:

**Definition 1.** *Let $\mathcal{D}$ denote some empirical distribution over a dataset D. Then we say that some feature $i \in [n]$ is* redundant *with respect to $\mathcal{D}$ if for any pair $\boldsymbol{x}, \boldsymbol{z} \in D$ it holds that $f(\boldsymbol{x}_{[n]\setminus\{i\}}; \boldsymbol{z}_{\{i\}}) = f(\boldsymbol{x})$.*

Here, the notation $f(\mathbf{x}_{[n]\setminus\{i\}}; \mathbf{z}_{\{i\}})$ indicates that all features in $[n] \setminus \{i\}$ are fixed to their values in $\mathbf{x}$, while feature $i$ is set to its value in $\mathbf{z}$. Notably, this is equivalent to defining:

$$\forall S \subseteq [n], \mathbf{z} \in D \quad f(\mathbf{x}_S; \mathbf{z}_{\bar{S}}) = f(\mathbf{x}_{S\setminus\{i\}}; \mathbf{z}_{\bar{S}\cup\{i\}}) \tag{68}$$

As before, the notation $(\mathbf{x}_S; \mathbf{z}_{\bar{S}})$ indicates that the features in $S$ are fixed to their values in $\mathbf{x}$, while the features in $\overline{S}$ are fixed to their values in $\mathbf{z}$. Thus, in this sense, if we "remove" feature $i$ from the input space and define a new function $f'$ over the reduced space $[n'] := [n] \setminus \{i\}$, then for any input of size $n'$, the output of $f'$ will exactly match the output of $f$ when applied to the same features $[n] \setminus \{i\}$.

This removal can be done in polynomial time for both tree ensembles and neural networks: for neural networks, it involves detaching the corresponding input neuron from the network; for tree ensembles, it involves removing any splits on that feature from all decision trees. Given the empirical nature of the contrastive value function $v_{con}^g(S)$ — as previously discussed in this proof and in Proposition 2 — we can compute each $v_{con}^g(\{i\})$ for all $i \in [n]$ by iterating over all pairs $\mathbf{x}, \mathbf{z} \in \mathcal{D}$ and checking whether:

$$f(\mathbf{x}_{[n]\setminus\{i\}}; \mathbf{z}_{\{i\}}) \neq f(\mathbf{x}) \tag{69}$$

If this condition holds, it indicates that feature $i$ is not redundant. Conversely, if the condition fails for all pairs considered during iteration, then $i$ is deemed redundant. Once all redundant features have been identified with respect to the empirical distribution $\mathcal{D}$, we can remove them and construct an equivalent model $f'$. As discussed above, this transformation can be performed in polynomial time for both neural networks and decision trees.

To conclude our proof, we observe that $v_{con}^g(\{i\})$ equals $0$ for some empirical distribution $\mathcal{D}$ if and only if feature $i$ is redundant with respect to $\mathcal{D}$. Therefore, once the preprocessing step removes all redundant features from $f$ and we construct the resulting function $f'$, we ensure that $\min_{i \in [n]} v_{con}^g(\{i\}) > 0$.

We now present the more precise bounds referenced in our proof. Specifically, we demonstrate that the approximation factor is $\mathcal{O}(\ln(|\mathbf{D}|))$. Given that the empirical distribution D is fixed, this yields a constant approximation. While tighter bounds may be achievable, our goal here is solely to establish that the bound is *constant* — a property that will later sharply contrast with the local explainability setting, where no bounded approximation exists.

More specifically, we know that the probability function $\mathbf{Pr}_{\mathbf{z} \sim \mathcal{D}}(f(\mathbf{x}) \neq f(\mathbf{z}) \mid \mathbf{x}_{\bar{S}} = \mathbf{z}_{\bar{S}})$ is, by the definition of empirical distributions, at least $\frac{1}{|\mathbf{D}|}$. Therefore, the definition of $\mathbb{E}_{\mathbf{x} \sim \mathcal{D}}[\mathbf{Pr}_{\mathbf{z} \sim \mathcal{D}}(f(\mathbf{x}) \neq f(\mathbf{z}) \mid \mathbf{x}_{\bar{S}} = \mathbf{z}_{\bar{S}})]$ (i.e., the value function $v_{con}^g$) is at least $\frac{1}{|\mathbf{D}|^2}$. In particular, this also implies that:

$$\min_{i \in [n]} v_{con}^g(\{i\}) \geq \frac{1}{|\mathbf{D}|^2} \tag{70}$$

Consequently, we obtain that:

$$\ln\left(\frac{v_{con}^g([n])}{\min_{i \in [n]} v_{con}^g(\{i\})}\right) \leq \ln(|\mathbf{D}|^2) = \mathcal{O}(\ln(|\mathbf{D}|)) \tag{71}$$

Which concludes our proof.

$\square$

**Lemma 11.** *Unless PTIME=NP, there is no bounded approximation for computing a cardinally minimal local $\delta$-contrastive reason for any $\langle f, \mathbf{x} \rangle$ where $f$ is either a neural network or a tree ensemble, even when $|D| = 1$.*

*Proof.* Assume, for contradiction, that there exists a bounded approximation algorithm for computing a cardinally minimal local $\delta$-contrastive reason for some $\langle f, \mathbf{x} \rangle$, where $f$ is a neural network or a tree ensemble—even when $|\mathbf{D}| = 1$. However, Lemma 9 establishes that even with a single baseline $\mathbf{z} = 0_n$ (i.e., when the entire dataset is just one instance), deciding whether a contrastive reason exists is NP-hard unless PTIME = NP. Therefore, if a polynomial-time algorithm could yield a bounded approximation for a cardinally minimal contrastive reason, it would contradict this result, as such an approximation would implicitly decide the existence of a contrastive explanation.

$\square$

## L  PROOF OF THEOREM 4

**Theorem 4.** *Given a neural network or tree ensemble $f$ and an empirical distribution $\mathcal{D}$ over a fixed dataset D, Algorithm 2 yields a* constant $\mathcal{O}\left(\frac{1}{1-k^f} + \ln\left(\frac{v_{suff}^g([n])}{\min_{i \in [n]} v_{suff}^g(\{i\})}\right)\right)$*-approximation for*

*computing a global cardinally minimal $\delta$-sufficient reason for $f$, assuming feature independence. In contrast, unless PTIME=NP, there is no bounded approximation for computing a local cardinally minimal $\delta$-sufficient reason for any $\langle f, \boldsymbol{x} \rangle$, even when $|\boldsymbol{D}| = 1$.*

*Proof.* We divide the proof into two lemmas: one establishing the approximation guarantee for the global case, and the other demonstrating the intractability of the local case.

**Lemma 12.** *Given a neural network or tree ensemble $f$ and an empirical distribution $\mathcal{D}$ over a fixed dataset D, Algorithm 2 yields a constant $\mathcal{O}\left(\frac{1}{1-k^f} + \ln\left(\frac{v_{suff}^g([n])}{\min_{i \in [n]} v_{suff}^g(\{i\})}\right)\right)$-approximation for computing a global cardinally minimal $\delta$-sufficient reason for $f$, assuming feature independence.*

*Proof.* The proof will follow a similar approach to that of Lemma 10, where we showed that for both neural networks and tree ensembles, Algorithm 2 achieves an approximation factor of $\ln\left(\frac{v_{con}^g([n])}{\min_{i \in [n]} v_{con}^g(\{i\})}\right)$. After applying the preprocessing step, this ratio is guaranteed to be finite and bounded by $\mathcal{O}(\ln(|\mathbf{D}|))$.

Similar to Lemma 10, we will again prove a stronger claim — namely, that the result holds for *any* model, assuming the trivial condition that its inference time is polynomially computable, and the additional condition concerning the removal of redundant features, as described in Lemma 10. As discussed there, both conditions are satisfied by our neural network and tree ensemble formalizations.

Here as well, since we are working with empirical distributions, the computation of the global sufficient probability value function:

$$v_{\text{suff}}^g(S) = \mathbb{E}_{\mathbf{x} \sim \mathcal{D}}[\mathbf{Pr}_{\mathbf{z} \sim \mathcal{D}}(f(\mathbf{x}) = f(\mathbf{z}) \mid \mathbf{x}_S = \mathbf{z}_S)] \tag{72}$$

can be computed in polynomial time by iterating over all pairs $\mathbf{x}, \mathbf{z}$ in the dataset D, as previously shown in both Lemma 10 and Proposition 2. Since Algorithm 2 makes only a linear number of such polynomial-time queries, its overall runtime is polynomial.

Unlike Lemma 10, where the approximation relied on the result by Wolsey et al. (Wolsey, 1982) for monotone non-decreasing submodular functions, the setting here requires a different condition due to the supermodular nature of the function. Specifically, Shi et al.(Shi et al., 2021) provided an approximation guarantee of $\mathcal{O}\left(\frac{1}{1-k^f} + \ln\left(\frac{v_{\text{suff}}^g([n])}{\min_{i \in [n]} v_{\text{suff}}^g(\{i\})}\right)\right)$, where $k^f := 1 - \min_{i \in [n]} \frac{v([n]) - v([n] \setminus i)}{v(i) - v(\emptyset)}$. We established in Proposition 2 that the value function $v_{\text{suff}}^g$ is monotone non-decreasing, and under the assumption of feature independence, Proposition 3 further shows that it is supermodular. Given that Algorithm 2 runs in polynomial time, this leads directly to an approximation guarantee of $\mathcal{O}\left(\frac{1}{1-k^f} + \ln\left(\frac{v_{\text{suff}}^g([n])}{\min_{i \in [n]} v_{\text{suff}}^g(\{i\})}\right)\right)$, where $k^f := 1 - \min_{i \in [n]} \frac{v_{\text{suff}}^g([n]) - v_{\text{suff}}^g(([n] \setminus \{i\}))}{v_{\text{suff}}^g(\{i\}) - v_{\text{suff}}^g(\emptyset)}$.

To show that this expression is both bounded and constant, we follow the same preprocessing step as in Lemma 10, where we remove all redundant features from $f$ — a process that, as previously explained, can be carried out in polynomial time for both neural networks and tree ensembles. As before, a feature $i$ is strictly redundant if and only if $v(\{i\}) - v(\emptyset) = 0$. This preprocessing yields a new function $f'$ that behaves identically to $f$ over the remaining features and ensures that, for every $i$, both $v_{\text{suff}}^g(\{i\}) > 0$ and $v_{\text{suff}}^g(\{i\}) - v_{\text{suff}}^g(\emptyset) > 0$ hold.

In order for us to show that this expression is both bounded and constant, similarly to Lemma 10 we will perform the exact same preprocessing phase (which, as wel explained there, can be performed in polynomial time both for neural networks as well as tree ensembles) as before where we remove all redundant features from $f$. We note that here too, it holds that a feature $i$ is strictly redundant if and only if $v(\{i\}) - v(\emptyset) = 0$. Hence, this preprocessing phase will give us a new function $f'$ for which it both holds that it is equivalent to the behaviour of $f$ for all remaining feautres and also satisfies that for any $i$ it holds that both $v_{\text{suff}}^g(\{i\}) > 0$ and $v_{\text{suff}}^g(\{i\}) - v_{\text{suff}}^g(\emptyset) > 0$ hold.

Now, following the same reasoning as in Lemma 10, where the probability term is lower bounded by $\frac{1}{|\mathbf{D}|}$ and its expected value by $\frac{1}{|\mathbf{D}|^2}$, we obtain that $\min_{i \in [n]} v_{\text{suff}}^g(i) \geq \frac{1}{|\mathbf{D}|^2}$ and $v_{\text{suff}}^g(i) - v_{\text{suff}}^g(\emptyset) \geq \frac{1}{|\mathbf{D}|^2}$. This implies, as in Lemma 10, that the first term is lower bounded by $\mathcal{O}(\ln(|\mathbf{D}|))$, and for the second term we derive the following:

Since $v_{\text{suff}}^g$ is supermodular (Proposition 3), we have that $v_{\text{suff}}^g([n]) - v_{\text{suff}}^g(([n] \setminus \{i\})$. It follows that $1 \leq v_{\text{suff}}^g([n]) - v_{\text{suff}}^g([n] \setminus \{i\}) \leq \frac{1}{|D|^2}$. This implies: $1 \leq \min_{i \in [n]} \frac{v_{\text{suff}}^g([n]) - v_{\text{suff}}^g([n] \setminus \{i\})}{v_{\text{suff}}^g(\{i\}) - v_{\text{suff}}^g(\emptyset)} \leq |D|^2$, and therefore, $1 - |D|^2 \leq k^f \leq 0$. This yields $\frac{1}{1-k^f} \leq |D|^2$, demonstrating that the overall approximation bound based on the one established by Shi et al. is constant. While this bound may be significantly smaller in practice, our goal here is simply to show that it remains constant — unlike in the local setting, where no bounded approximation is achievable.

$\square$

**Lemma 13.** *Unless PTIME=NP, there is* no bounded approximation *for computing a local cardinally minimal $\delta$-sufficient reason for any $\langle f, \boldsymbol{x} \rangle$, even when $|D| = 1$.*

*Proof.* The proof follows a similar approach to Lemma 11. Suppose, for contradiction, that there exists a polynomial-time algorithm that provides a bounded approximation to a cardinally minimal local $\delta$-sufficient reason for some $\langle f, \mathbf{x} \rangle$, where $f$ is either a neural network or a tree ensemble—even when $|D| = 1$. Yet, Lemma 6 shows that even in the extreme case where the dataset consists of a single baseline $\mathbf{z} = 0_n$, deciding whether a sufficient reason exists is NP-hard unless PTIME = NP. Hence, the existence of such an approximation algorithm would contradict this hardness result, as it would entail the ability to decide the existence of a sufficient reason.

$\square$

## M  HARDNESS OF APPROXIMATING GLOBAL CONTRASTIVE REASONS FOR NEURAL NETWORKS USING EMPIRICAL DISTRIBUTIONS

**Proposition 1.** *Assume a neural network $f : \mathbb{F} \to [c]$ with ReLU activations, some dataset $\mathbf{D}$, such that $\mathcal{D}$ is the empirical distribution defined over $\mathbf{D}$, and some threshold $\delta \in \mathbb{Q}$. Then unless $NP \subseteq DTIME(n^{\mathcal{O}(log(log(n)))})$, there does not exist a polynomial-time $\mathcal{O}(\ln(|\mathbf{D}|) - \epsilon)$ approximation algorithm for computing a cardinally minimal global contrastive reason for $f$ concerning $\mathcal{D}$, for any fixed $\epsilon > 0$.*

*Proof.* The proof proceeds via a polynomial-time approximation-preserving reduction from a specific variant of Set Cover that is, similarly to regular Set Cover, provably hard to approximate within the factor given in the proposition (Vazirani, 2001).

---

**Partial Set Cover**:
**Input**: A universe $U := \{u_1, \ldots, u_n\}$, a collection of sets $S := \{S_1, \ldots S_m\}$ such that for all $j$ $S_j \subseteq U$, and a coverage threshold $\delta \in \{0, 1, \ldots, n\}$.
**Output**: A minimum-size subset $C \subseteq S$ such that $|\cup_{S_j \in C} S_j| \geq \delta$.

---

We reduce this problem to that of computing a *global*, cardinally-minimal contrastive reason $S$ for a neural network $f$ with respect to a distribution $\mathcal{D}$ defined over a dataset $\mathbf{D}$, and some threshold $\delta'$. We begin by recalling the definition of a global contrastive reason and then extend it to the setting where the explanation is evaluated over an empirical dataset $\mathbf{D}$. The definition is as follows:

$$\mathbb{E}_{\mathbf{x} \sim \mathcal{D}}[\mathbf{Pr}_{\mathbf{z} \sim \mathcal{D}}(f(\mathbf{z}) \neq f(\mathbf{x}) \mid \mathbf{z}_{\bar{S}} = \mathbf{x}_{\bar{S}})] \geq \delta. \tag{73}$$

More specifically, when we assume an empirical distribution over $\mathbf{D}$, this implies that:

$$\mathbb{E}_{\mathbf{x} \sim \mathcal{D}}[\mathbf{Pr}_{\mathbf{z} \sim \mathcal{D}}(f(\mathbf{z}) \neq f(\mathbf{x}) \mid \mathbf{z}_{\bar{S}} = \mathbf{x}_{\bar{S}})] =$$
$$\frac{1}{|\mathbf{D}|} \sum_{\mathbf{x} \in \mathbf{D}} [\mathbf{Pr}_{\mathbf{z} \sim \mathcal{D}}(f(\mathbf{z}) \neq f(\mathbf{x}) \mid \mathbf{z}_{\bar{S}} = \mathbf{x}_{\bar{S}})] =$$
$$\frac{1}{|\mathbf{D}|} \sum_{\mathbf{x} \in \mathbf{D}} \left( \frac{\sum_{\mathbf{z} \in \mathbf{D}_{\bar{S}}(\mathbf{z}, \mathbf{x})} \mathbf{1}_{\{f(\mathbf{z}) \neq f(\mathbf{x})\}}}{\mathbf{D}_{\bar{S}}(\mathbf{z}, \mathbf{x})} \right) \tag{74}$$

where we denote $\mathbf{D}_{\bar{S}}(\mathbf{z}, \mathbf{x}) := \{\mathbf{z} \in \mathbf{D} : \mathbf{z}_{\bar{S}} = \mathbf{x}_{\bar{S}}\}$. We now construct a ReLU neural network $f$. Before doing so, we specify the functional form that we intend the constructed neural network $f$ to

implement. For each set $S_j$, we introduce two vectors $\mathbf{x}^j$ and $\mathbf{y}^j$, both belonging to $\mathbb{Z}^n$ and drawn from the dataset $\mathbf{D}$, such that they agree on all coordinates outside $S_j$; that is, they share identical feature values on the complement $\bar{S}_j$. Formally:

$$\mathbf{x}^j_{\bar{S}_j} = \mathbf{y}^j_{\bar{S}_j} \tag{75}$$

and we assign strictly different values to the features in $S_j$ by selecting them from two distinct vectors $\mathbf{z}$ and $\mathbf{z}'$ such that:

$$\mathbf{x}^j = (\mathbf{x}^j_{\bar{S}_j}; \mathbf{z}^j_{S_j}) \neq (\mathbf{x}^j_{\bar{S}_j}; \mathbf{z}'^j_{S_j}) = (\mathbf{y}^j_{\bar{S}_j}; \mathbf{z}'^j_{S_j}) = \mathbf{y}^j \tag{76}$$

We aim to construct $f$ such that it satisfies the following:

$$f(\mathbf{x}^j) = f(\mathbf{x}^j_{\bar{S}_j}; \mathbf{z}^j_{S_j}) \neq f(\mathbf{x}^j_{\bar{S}_j}; \mathbf{z}'^j_{S_j}) = f(\mathbf{y}^j_{\bar{S}_j}; \mathbf{z}'^j_{S_j}) = f(\mathbf{y}^j) \tag{77}$$

Let us now assume that both $\mathbf{x}^j$ and $\mathbf{y}^j$ occur *uniquely* in the dataset $\mathbf{D}$ across all $[n]$ coordinates — that is, no other vector in $\mathbf{D}$ matches either of them on any coordinate. Under this assumption, we obtain the following relation:

$$\mathbb{E}_{\mathbf{x}\sim\mathcal{D}}[\mathbf{Pr}_{\mathbf{z}\sim\mathcal{D}}(f(\mathbf{z}) \neq f(\mathbf{x}) \mid \mathbf{z}_{\bar{S}_j} = \mathbf{x}_{\bar{S}_j})] =$$
$$\frac{1}{|\mathbf{D}|} \sum_{\mathbf{x}\in\mathbf{D}} \left( \frac{\sum_{\mathbf{z}\in\mathbf{D}_{\bar{S}_j}(\mathbf{z},\mathbf{x})} \mathbf{1}_{\{f(\mathbf{z})\neq f(\mathbf{x})\}}}{\mathbf{D}_{\bar{S}_j}(\mathbf{z},\mathbf{x})} \right) = \frac{2}{2|\mathbf{D}|} = \frac{1}{|\mathbf{D}|} \tag{78}$$

Here, the factor of 2 appearing in both the numerator and denominator arises from the fact that only two vector pairs — $(\mathbf{x}^j, \mathbf{z}^j)$ and $(\mathbf{z}^j, \mathbf{x}^j)$ — contribute to the respective sums.

We repeat this construction for each $S_i$ in the set-cover instance, each time choosing two *distinct* vectors $\mathbf{x}^i$ and $\mathbf{y}^i$ that differ on *every* coordinate in $[n]$. In total, this yields $2m$ vectors that form the dataset $\mathbf{D}$ from which we construct $f$.

We then construct a neural network $f$ that simulates this function. This is achieved using a known result on *neural network memorization* (Vardi et al., 2022), which guarantees that a ReLU network capable of representing any finite set of input–output pairs can be built in polynomial time. Using this result, we construct $f$ accordingly, and set $\delta' := \frac{\delta}{|\mathbf{D}|}$.

We now argue that a subset $S_j$ is a global contrastive reason for $f$ under tolerance $\delta'$ if and only if $C$ is a partial set cover for $U$ under tolerance $\delta$. By Equation 78, each constructed pair of vectors $(\mathbf{x}^j, \mathbf{y}^j)$ contributes exactly $\frac{1}{|\mathbf{D}|}$ to the value term. Moreover, because each $S_j$ uses its own distinct pair of vectors in $\mathbb{Z}^n$, there is no overlap between these pairs. Consequently, the coverage achieved by any set cover $C$ corresponds exactly to the accumulated contribution of the selected subsets in the input domain of $f$. This completes the reduction.

$\square$

## N  TRACTABILITY RESULTS FOR ORTHOGONAL DNFS

**Lemma 14.** *If $f$ is an orthogonal DNFs and the probability term $v^g_{suff}$ can be computed in polynomial time given the distribution $\mathcal{D}$ (which holds for independent distributions, among others), then obtaining a subset-minimal global $\delta$-sufficient reason can be obtained in polynomial time.*

*Proof.* We begin by defining orthogonal DNFs (Crama & Hammer, 2011) before presenting the full proof.

**Definition 2** (Orthogonal DNFs). *A Boolean function $\varphi$ is in* orthogonal disjunctive normal form *(orthogonal DNF) (Crama & Hammer, 2011) if it is a disjunction of terms $T_1 \vee T_2 \vee \cdots \vee T_m$, such that for every pair of distinct terms $T_i$ and $T_j$ ($i \neq j$):*

$$T_i \wedge T_j \models \bot$$

*This means that no single variable assignment can satisfy more than one term simultaneously.*

The proof mirrors the argument used in Theorem 1, relying on the fact that $v_{\text{suff}}^g$ can be computed in polynomial time for any set $S$.

For a given set $S$, let $\mathcal{C}^S T_i$ be the constraints that the term $T_i$ imposes on the variables in $S$. We enumerate all pairs of terms $(T_i, T_j)$. Any pair whose constraints on $S$ are *inconsistent* (i.e., $\mathcal{C}_{T_i}^S \wedge \mathcal{C}_{T_j}^S \models \bot$) is ignored, since it contributes zero to $v^g$suff.

For each *consistent* pair of terms $(T_i, T_j)$, we evaluate $\mathbf{Pr}(\mathbf{x}_S \in \mathcal{C}_{T_i,T_j}^S)$, where $\mathcal{C}_{T_i,T_j}^S = \mathcal{C}_{T_i}^S \cap \mathcal{C}_{T_j}^S$. The contribution of this pair to $v_{\text{suff}}^g$ is then

$$\mathbf{Pr}(\mathbf{x}_S \in \mathcal{C}_{T_i,T_j}^S) \cdot \mathbf{Pr}(\mathbf{x}_{\bar{S}} \in \mathcal{C}_{T_i}^{\bar{S}}) \cdot \mathbf{Pr}(\mathbf{x}_{\bar{S}} \in \mathcal{C}_{T_j}^{\bar{S}}).$$

Under feature independence, these probabilities factor as $\Pr(\mathbf{x}_S) = \prod_{i \in S} p(x_i)$. Since there are at most $m^2$ such term pairs, the value of $v_{\text{suff}}^g$ can be computed in polynomial time under an independent distribution $\mathcal{D}$. $\qquad\square$

## O    DISCLOSURE: USAGE OF LLMS

An LLM was used exclusively as a writing assistant to refine grammar and typos and improve clarity. It did not contribute to the generation of research ideas, study design, data analysis, or interpretation of results, all of which were carried out solely by the authors.

