# OpenReview forum: "Unifying Formal Explanations: A Complexity-Theoretic Perspective"
_ICLR.cc/2026/Conference — ICLR 2026 Poster_

### Official Review · Reviewer_cPWT · 2025-10-18

**Soundness:** 2
**Presentation:** 3
**Contribution:** 2
**Rating:** 4
**Confidence:** 4

**Summary:**

This paper presents new findings in the field of formal explainable artificial intelligence (XAI), focusing on the complexity of sufficient and contrastive explanations, including their probabilistic variants. The authors introduce a unified framework that covers both local and global explanations, providing insights into the complexity and approximability of subset-minimal and cardinality-minimal explanations.

The key takeaway from this study is that global probabilistic explanation problems differ significantly from local ones in terms of computational complexity and approximability. Specifically, the objective function for both sufficient and contrastive global probabilistic explanations is monotone increasing (Proposition 2). Additionally, when dealing with joint distributions, the objective function exhibits supermodularity for sufficient global probabilistic explanations (Proposition 3) and submodularity for contrastive global probabilistic explanations (Proposition 4). In contrast, local probabilistic explanations lack these properties.

The authors also focus on empirical distributions, where the objective function can be evaluated in polynomial time for various classifiers. They combine the aforementioned properties with standard greedy algorithms to reveal new results. Notably, subset-minimal global explanations can be computed efficiently for empirical distributions (Theorem 2), and cardinality-minimal global explanations are approximable (up to curvature and a logarithmic term) for empirical joint distributions (Theorems 3 and 4).

**Strengths:**

**S1.** Despite the paper's dense and technical nature, it is well-articulated: the notation is clear, and the results are easy to understand.

**S2.** The related work is well-detailed, including most relevant references.

**S3.** The unified framework that encompasses both local and global explanations, including probabilistic approaches, is intuitive.

**S4.** To my knowledge, Propositions 2, 3, and 4 appear to be novel contributions. In most cases, the negative results are not straightforward.

**Weaknesses:**

**W1.** I find Algorithm 2 confusing, as it seems incorrect; the marginal gain should be maximized.

**W2.** I think that the positive results presented in the main theorems (1-4) are primarily applications of existing findings.

**W3.** Theorem 4 does not truly deliver a constant-factor approximation, as the curvature could be unbounded.

**W4.** The upper bounds for Theorems 3 and 4 appear somewhat loose, and these approximation results lack lower bounds necessary to establish tightness.

**W5.** Finally, the practicality of Theorems 3 and 4 is questionable, given that empirical “joint” distributions are rarely available in real-world scenarios.

**Questions:**

------------------------------
### Major Comments:
------------------------------

**C1**. As mentioned earlier, I found Algorithm 2 to be confusing. Typically, a greedy algorithm aims to maximize the marginal gain at each step, which is especially true for the greedy set-cover method. Therefore, unless I missed something, I believe that in Line 3, the `argmin` function should be replaced with an `argmax` function. Consequently, the comment in Line 428 should also be revised accordingly.

**C2**. Essentially, the approximation factor for Theorem 3 consists of a logarithmic term (related to the number of data instances), while the approximation factor for Theorem 4 includes both a logarithmic term and a curvature term. Can we bound the curvature term? If not, the fourth contribution mentioned in the introduction (Lines 120-126) should be rephrased to reflect this.

**C3**. In light of the previous comment, could the authors provide tight lower bounds for Theorems 3 and 4? Specifically, for large datasets, a bound of the form $\log |D|$ in Theorem 3 seems quite loose. However, if the authors demonstrate that approximating the problem within a factor of $(1 - \epsilon) \ln |D|$ is NP-hard, this would strengthen the result. A similar observation applies to Theorem 4, given that the curvature can be significant.

**C4**. In Theorems 2-4, the assumption regarding “empirical” distributions is reasonable; otherwise, the problem could be PP-hard. However, combining this assumption with the condition that $\mathcal D$ is also a “joint” distribution appears unrealistic in practice. Do the authors envision practical scenarios where we need to globally explain a model trained on a dataset with samples drawn from a joint distribution? At present, I find this result to be primarily of theoretical interest, as I have not identified concrete applications for it.

-----------------
### Minor Comments:
------------------

**C5**. In Section 2 (Setting), I suggest introducing the main notation, including the input dimension ($n$), the number of classes ($c$), the underlying distribution ($\mathcal D$), and the training set ($\mathbf D$ or $\mathbf Z$). Additionally, I would like to ask why you are using the subscript $p$ in $\mathcal D_p$; I would recommend simply using $\mathcal D$. Throughout the rest of the paper, I suggest maintaining consistent notation (for example, choose either P or PTIME, but not both).

**C6**. Based on the proof provided, I believe that Theorem 1 applies not only to decision trees but also to orthogonal DNF formulas (i.e., "1-satisfy" DNF formulas). Given that there exists a Fully Polynomial Randomized Approximation Scheme for counting models of arbitrary DNF formulas, I am also curious whether we could find an approximation result for identifying minimal subset global explanations under the uniform distribution.

**C7**. Section 8 is acceptable, but it could be more informative. I recommend including some key open questions that arise from this theoretical study. For instance, can we establish better approximation bounds for minimal-size global explanations when the classifier is a simple function, such as a decision tree or a linear threshold function?

---

> ### Author Response · Authors · 2025-11-27
>
> We thank the reviewer for their especially thorough review, the many insightful comments and constructive suggestions, and for recognizing the importance of our work. Our detailed response appears below.
>
>
> **A typo in Algorithm 2**
>
> We thank the reviewer for catching this important point! This was indeed a typo - the algorithm should of course use *argmax*, as is standard in set-cover–related problems. The main distinction between Algorithm 1 and Algorithm 2 lies in whether they take a bottom-up or top-down approach, which leads to different guarantees. We have corrected this in the revised manuscript and appreciate the reviewer for bringing this to our attention.
>
> **Is the curvature term in the approximation factor of Theorem 2 bounded?**
>
> Yes, as discussed in Appendix K, the curvature term *is bounded* and satisfies an upper bound of $|\mathbf{D}|^2$. While this bound is indeed significantly looser compared to the global contrastive case, it still represents a substantial improvement over the *local* variant, which admits *no bounded* approximation - even under the extreme assumption $|\mathbf{D}| = 1$. This is precisely the distinction we aimed to highlight. Intuitively, any attempt to formulate an analogous curvature expression for the local setting inevitably leads to a denominator of zero, making it impossible for the bound to remain finite.
>
> More specifically, the difficulty in obtaining a good approximation for the global *sufficient* case, compared to the global *contrastive* case, stems from the different underlying combinatorial structures. The sufficient case reduces to a *submodular set cover* problem, for which well-known approximation guarantees exist. In contrast, the global contrastive case corresponds to a *supermodular set cover* problem, which is widely regarded as significantly harder to approximate due to the increasing marginal gains inherent to supermodular functions.
>
> We believe this highlights an interesting, clean and rich hierarchy revealed by our results: (1) *Global contrastive* reasons offer the strongest guarantees, thanks to their monotone and *submodular* structure. (2) *Global sufficient* reasons yield weaker, yet still meaningful guarantees, reflecting their monotone *supermodular* nature. (3) The *local* setting (both sufficient and contrastive) is the most challenging: here, neither monotonicity nor (sub/super)modularity holds, and no bounded approximation can be obtained.
>
> **Can Theorem 1 be extended to Orthogonal DNF formulas and other tractable models?**
>
> Yes - the positive result of Theorem 1, showing that global subset-minimal explanations can be computed in polynomial time, carries over directly to orthogonal DNFs. Their pairwise-inconsistent terms behave analogously to the root-to-leaf paths of a decision tree, making the extension immediate. We agree with the reviewer that this generalization is valuable to highlight. Accordingly, we have added a formal proof in Appendix M and now reference it in the main text (lines 403–408) immediately following Theorem 1.
>
> Moreover, we note that this positive result extends to additional classes of structurally decomposable and deterministic Boolean circuits, including SDDs and OBDDs. For these representations, one can compile the value function $v_{suff}^g$ in time polynomial in the size of the circuit for any given set $S$. Assuming feature independence, the resulting circuit supports weighted model counting, allowing us to compute the numerical value of $v^{g}_{suff}$. We will elaborate on these generalizations in the final version and thank the reviewer for highlighting this valuable direction.

---

> ### Author Response · Authors · 2025-11-27
>
> **Reliance on previous combinatorial optimization findings for positive complexity results**
>
> Our work introduces a novel and unique connection between the foundational explainability tasks of identifying sufficient and contrastive reasons and fundamental concepts in combinatorial optimization, such as monotonicity, submodularity, and supermodularity. We believe that this novel connection between two distinct fields can pave the way for a broad range of future research that can be based on this established connection. By framing these explainability problems as combinatorial optimization tasks and by leveraging these novel structural properties, we are indeed able to apply previously studied greedy algorithms from the optimization literature with provable guarantees. However, and interestingly, our work’s results reveal that applying the *exact same algorithms* to different explainability settings (local or global explainability settings) leads to *strikingly different* theoretical guarantees. We view this as an important contribution to the explainability literature.
>
> Moreover, from a more technical standpoint, we emphasize that proving the monotonicity, submodularity, and supermodularity properties of the underlying value functions is non-trivial and relies on careful mathematical constructions (see Propositions 2, 3, and 4). Importantly, these results hold under very general conditions — across any model with polynomial-time inference, a broad class of distributions, and both discrete and continuous input domains. While the *approximation* guarantees of our algorithms are indeed drawn from established results from prior optimization literature, our *hardness* results, as the reviewer has also acknowledged, are entirely independent of those and are derived through carefully designed, non-trivial complexity reductions. These technical proofs enable us to demonstrate the distinction between local and global explanations by showing that the intractability observed in the local explainability setting (unlike in the global case) persists even under substantially simplified assumptions.
>
> **On the tightness of the approximation bounds in Theorem 3 and 4**
>
> We thank the reviewer for raising this insightful point. While $\mathbf{D}$ may be large in principle, many standard XAI methods do not use the full dataset when computing expectations. For computational reasons, many methods typically rely on a small sampled subset - for example, the SHAP library often uses only $\sim100$ samples, and some variants (e.g., baseline SHAP [1]) even use a single point. Thus, in practice, $\ln(|\mathbf{D}|)$ may, in some cases, be quite small. Moreover, the key message we wish to emphasize is the sharp *contrast* between the global setting, where such bounds hold, and the local setting, where even for a single datapoint no bounded approximation is possible. We believe this distinction is crucial for the XAI community in understanding which explanations are practically attainable.
>
> However, we agree that understanding the exact tightness of these bounds is important. Following the reviewer’s suggestion, we added to the appendix a proof showing that unless $\text{NP}\subseteq \text{DTIME}(n^{\mathcal{O}(\log(\log n))})$, no $\ln(|\mathbf{D}|)-\epsilon$ approximation exists for computing a cardinally-minimal global contrastive reason under an empirical distribution $\mathbf{D}$ for neural networks, via a reduction from *partial set cover*. This proof appears in Appendix L and will be incorporated into the final version. Notably, this reduction does not assume feature independence. Determining whether a comparable lower bound holds under independence, or whether the bound under the independence setting can be tightened, remains an interesting open question.

---

> ### Author Response · Authors · 2025-11-27
>
> **The practicality of the feature independence assumption**
>
> We agree that feature independence is an “idealized’’ assumption. Still, it is widely used in many theoretical XAI works (e.g., 2–5) and helps cleanly identify which guarantees hold under which conditions. For example, *monotonicity* does *not* rely on independence and holds broadly for global sufficient and contrastive reasons under arbitrary (discrete or continuous) distributions. By contrast, submodularity and supermodularity *do* depend on independence, and as shown in Appendix E, both fail once this assumption is removed. This distinction clarifies when tractability can, and cannot, be ensured.
>
> Regarding empirical distributions, we agree that real data is often correlated; however, our results still capture several meaningful scenarios studied in prior work. For example, consider a feature space $\mathbb{F}:=\lbrace 0,1\rbrace^n$ and a dataset $\mathbf{D}$ built from a point $\mathbf{x}$ (or a small set $\lbrace\mathbf{x}^1,\ldots,\mathbf{x}^d\rbrace$) together with an augmentation of *all* perturbations of Hamming radius at most $k$ over some subset $S\subseteq[n]$ or a small collection of such subsets. This setting of modifying up to $k$ features in every possible way is standard in work on stability and sufficiency  (e.g., [6,7], where complementary features may vary within a radius $r$), and is closely related to bounded-sparsity counterfactuals (e.g., [8–9]). Since all perturbations up to radius $k$ are included, the induced distribution satisfies independence (and allows flexible density assignments). In this case, our $\mathcal{O}(\ln|\mathbf{D}|)$ bound becomes an $\mathcal{O}(k\cdot \ln |S|)=\mathcal{O}(k\cdot\ln n)$ approximation, which remains meaningful when $k$ is small.
>
> We agree that extending these guarantees to more expressive datasets and distributions is an important open question. While Appendix E shows that removing feature independence breaks submodularity and supermodularity, it is still interesting to explore whether broader, more “structured’’ distributional assumptions that are less restrictive than full independence might preserve these properties. We view this as a promising direction for future work.
>
> **Questions and remarks regarding notation**
>
> We thank the reviewer for these helpful suggestions. We initially used the notation $\mathcal{D}_p$ to avoid confusion with the dataset $\mathbf{D}$, but we agree that the distinction is clear enough; we therefore now use $\mathcal{D}$ throughout the revised manuscript. We also agree on the importance of consistent terminology and have updated all relevant instances to use “PTIME’’ instead of “P”.
>
> Finally, as suggested, we revised the “Preliminaries’’ section to clearly define $\mathcal{D}$, $\mathbf{D}$, the number of classes, the input dimension, and other basic notation. We also expanded Section 8 with a more detailed discussion of open theoretical directions, including tightening results for simpler model classes, exploring approximation techniques for expectations (e.g., FPRASs or Monte Carlo), and extending our results to additional feature-importance losses. We agree that these additions improve the clarity and motivation of the work and thank the reviewer for these helpful recommendations!
>
>
>
> [1] The many Shapley Values for Model Explanation (Sundararajan et al., ICML 2020)
>
> [2] On the Complexity of SHAP-Score-Based Explanations: Tractability via Knowledge Compilation and Non-Approximability Results (Arenas et al., JMLR 2023)
>
> [3] The Computational Complexity of Understanding Binary Classifier Decisions (Waldchen et al., JAIR 2021)
>
> [4] Provably Efficient, Succinct, and Precise Explanations (Blanc et al., NeurIPS 2021)
>
> [5] On Computing Probabilistic Explanations for Decision Trees (Arenas et al., NeurIPS 2022)
>
> [6] Stability Guarantees for Feature Attributions with Multiplicative Smoothing (Xue et al., NeurIPS 2023)
>
> [7] Probabilistic Stability Guarantees for Feature Attributions (Jin et al., NeurIPS 2025)
>
> [8] A Query-Optimal Algorithm for Finding Counterfactuals (Blanc et al., ICML 2022)
>
> [9] Explaining Machine Learning Classifiers Through Diverse Counterfactual Explanations (Mothilal et al., FAT* 2020)

---

> > ### Comment · Reviewer_cPWT · 2025-11-27
> >
> > Thank you for your thorough responses! The revised version is significantly improved. Since most of my concerns have been addressed, I believe the paper's strengths outweigh its weaknesses. I have adjusted my review accordingly.
> >
> > In an extended version of this paper, I would suggest providing an empirical proof-of-concept that illustrates the behavior of the greedy method relative to the neighborhood distribution you mentioned in your comment.

---

### Official Review · Reviewer_uoPW · 2025-10-27

**Soundness:** 3
**Presentation:** 3
**Contribution:** 4
**Rating:** 8
**Confidence:** 4

**Summary:**

The paper proposes a framework for unifying two types of explanations: contrastive and sufficient reason explanations. The framework models both explanations as minimisations of a value function.

**Strengths:**

- Clear research question and contribution
- A lot of novel (formal) insights are generated, which I consider to be valuable for the XAI community.

**Weaknesses:**

- Readability and accessibility can be improved. Most importantly, it looks to me that sufficient reasons are the same as semi-factual explanation, and global contrastive reasoning seems to describe goup/multi-instance counterfactuals. Since semi and counterfactuals are popular and widely used terms in the XAI community, I suggest clearly relating them to the concepts introduced and discussed in this paper. By this, the paper and its contribution will become accessible to a wider audience.

Minor:
- Line 232 "smaller" in XAI people often talk about "simpler explanation". I suggest clarifying the meaning of "smaller" and also including "simpler" to make the paper more accessible to other researchers

**Questions:**

None

---

> ### Author Response · Authors · 2025-11-27
>
> We appreciate the reviewer’s constructive feedback and their recognition of the importance of our work. Our detailed response is provided below.
>
> **Relation to counterfactuals, semifactuals and group/multi-instance counterfactuals**
>
> We thank the reviewer for this important comment and agree that the terminology in the explainable AI literature often overlaps, making clarification important. In essence, semi-factual explanations indeed closely align with sufficient-reason explanations, while counterfactuals and algorithmic-recourse notions are close in meaning to contrastive-reason explanations. In addition, global explanations naturally relate to “group” explanations that apply to multiple instances.
>
> Following the reviewers’ remarks, we have revised our manuscript to already explicitly relate local sufficient reasons to semifactuals (Lines 185-188), local contrastive reasons to counterfactuals (Lines 201-203), and global contrastive reasons to group/multi-instance counterfactuals (Lines 207-208). In the final version, we plan to expand this further by adding to the appendix a dedicated section that provides formal definitions of these other established notions and clarifies how they connect to our framework.
>
> In particular, because the *probabilistic* forms of explanations that are the main focus of our study are defined with respect to an arbitrary distribution $\mathcal{D}$, they naturally encompass a very wide range of scenarios previously examined in the XAI literature. For example, the conclusions we draw from our proofs of monotonicity, submodularity, and supermodularity, as well as the associated tractability results, naturally extend to many of these previously studied notions. This observation, as the reviewer rightly notes, is important for improving accessibility to a wider audience. We thank the reviewer for raising this point and will make sure to address it clearly in the final version.
>
> **Simple explanations vs. minimal explanations**
>
> We thank the reviewer for raising this insightful point. We have revised line 232 to say “of smaller size” rather than “smaller”, to clearly indicate that we refer to minimality in terms of the *cardinality* of the explanation rather than its human-perceived simplicity. While both minimality and simplicity are desirable properties, they capture different aspects: minimality is connected to the faithfulness of the explanation and reflects *functional necessity* - for instance, identifying the *smallest possible set* of feature changes required to flip a loan decision - whereas simplicity reflects cognitive interpretability, i.e., how easily a human can understand the explanation.
>
> In some cases, the two notions can align - for example, a minimal set of feature changes that flips a model’s decision may also be simpler for a user to understand because it involves fewer components. However, they may also diverge: a pixel-level minimal sufficient or contrastive explanation might be extremely small yet visually opaque, whereas a larger, superpixel-based explanation (as commonly used in practical XAI pipelines) may be far more interpretable. In light of this helpful remark, we will add a discussion of this distinction in the final version of the manuscript, clarifying that our guarantees concern minimality and are complementary to, but do not necessarily capture, human-centric notions of simplicity.

---

> > ### Comment · Reviewer_uoPW · 2025-11-27
> >
> > Thanks for your rebuttal.

---

### Official Review · Reviewer_qstK · 2025-10-30

**Soundness:** 3
**Presentation:** 3
**Contribution:** 3
**Rating:** 6
**Confidence:** 2

**Summary:**

This paper proposes a unified framework for explanation complexity. Global value functions are monotone; under feature independence, sufficient variants are supermodular and contrastive variants submodular. This yeilds polynomial-time algorithms for subset-minimal global explanations and approximations for cardinality-minimal ones. Local explanations remain NP-hard.

**Strengths:**

The global versus local structural distinction appears to be new. Monotonicity holds without independence. Proofs are rigorous with counterexamples showing the necessity of assumptions.

**Weaknesses:**

Feature independence is required for approximation results. This limits practical applicability where features correlate.

**Questions:**

Could you clarify the candidate set specification in Algorithm 2?

---

> ### Author Response · Authors · 2025-11-27
>
> We appreciate the reviewer’s constructive feedback and their recognition of the importance of our work. Our detailed response is provided below.
>
>
> **The feature independence assumption for submodularity and supermodularity**
>
> We thank the reviewer for this important point. We would first like to emphasize, as the reviewer also noted, that the *monotonicity* property for both global sufficient and contrastive reasons *does not* rely on feature independence. It holds for *any arbitrary* distribution, whether defined over discrete or continuous inputs. This already yields very strong guarantees - most notably that computing *subset-minimal* global explanations is tractable and can be done in polynomial time even for complex models such as neural networks, in sharp contrast to the intractability of the corresponding *local* explanations.
>
> We agree that the stronger structural guarantees, i.e. *submodularity* in the global contrastive case and *supermodularity* in the global sufficient case, *do* require feature independence. As we demonstrate in Appendix E, these properties can break once independence is removed. We actually view this result as an interesting insight in its own right, since it clarifies the precise structural conditions under which computing explanations becomes tractable or intractable. For this reason, this “idealized’’ assumption of feature independence is *a standard assumption* in much of the theoretical XAI literature (e.g., 1–5): it isolates the structural effects of interest and makes tractability conditions easier to characterize. Moreover, this assumption also underlies many practical methods - most notably SHAP - where the conditional expectation is replaced by the marginal one [6], which is the version actually implemented in widely used SHAP libraries.
>
> Lastly, and importantly, we also recall the important result we have shown that in the *local* explanation setting, neither monotonicity nor submodularity or supermodularity hold, even under feature independence or even much stronger conditions. This further highlights the sharp divide between the local and global settings.
>
> We thank the reviewer for raising this point, and we will include a more detailed discussion of it within our final version.
>
>
> **A typo in Algorithm 2 and the candidate set specification**
>
> We believe that the reviewer’s question may stem from a small typo we originally had in Algorithm 2. Specifically, line 3 incorrectly used “argmin”, and we have now corrected this to “argmax” in the revised manuscript. Conceptually, Algorithm 2 is a standard bottom-up greedy procedure for combinatorial optimization that iteratively selects the element with the largest marginal gain until the threshold $\delta$ is reached. This stands in contrast to Algorithm 1, which performs an analogous process in a top-down direction. Although both algorithms rely on similar principles, they yield different guarantees and behave optimally under different conditions.
>
> [1] On the Complexity of SHAP-Score-Based Explanations: Tractability via Knowledge Compilation and Non-Approximability Results (Arenas et al., JMLR 2023)
>
> [2] The Computational Complexity of Understanding Binary Classifier Decisions (Waldchen et al., JAIR 2021)
>
> [3] Provably Efficient, Succinct, and Precise Explanations (Blanc et al., NeruIPS 2021)
>
> [4] On the tractability of SHAP explanations (Van den Broeck et al., JAIR 2022)
>
> [5] On Computing Probabilistic Explanations for Decision Trees (Arenas et al., NeurIPS 2022)
>
> [6] The many Shapley Values for Model Explanation (Sundararajan et al., ICML 2020)

---

### Meta-Review · Area_Chair_fwQL · 2025-12-30

**Summary:**

This paper provides a unifying framework for understanding complexity-theoretic results in global explanations.
There were some discussions on theoretical results and the underlying assumptions.

**Reviewer Concerns:**

Reviewer cPWT raised concerns on the novelty of the theoretical results and on the boundedness of approximation, which were addressed in the rebuttal.
Reviewer qstK and Reviewer uoPW did not raise crucial concerns.

**Reviewer Scores:**

Reviewer cPWT acknowledged the clarifications provided in the rebuttal, and thus it is likely that they would have increased their score to 5 or 6.
Reviewer qstK and Reviewer uoPW would likely have kept their scores unchanged.

---

### Decision · Program_Chairs · 2026-01-26

Accept (Poster)